# Defining cellular complexity in human autosomal dominant polycystic kidney disease by multimodal single cell analysis

Yoshiharu Muto [1], Eryn E. Dixon [1], Yasuhiro Yoshimura[1], Haojia Wu [1], Kohei Omachi[1], Nicolas Ledru [1], Parker C. Wilson [2], Andrew J. King[3], N. Eric Olson[3], Marvin G. Gunawan [4], Jay J. Kuo [4], Jennifer H. Cox[4], Jeffrey H. Miner [1], Stephen L. Seliger[5], Owen M. Woodward [6], Paul A. Welling[7], Terry J. Watnick[5] & Benjamin D. Humphreys [1,8] ✉

Autosomal dominant polycystic kidney disease (ADPKD) is the leading genetic cause of end stage renal disease characterized by progressive expansion of kidney cysts. To better understand the cell types and states driving ADPKD progression, we analyze eight ADPKD and five healthy human kidney samples, generating single cell multiomic atlas consisting of ~100,000 single nucleus transcriptomes and ~50,000 single nucleus epigenomes. Activation of proin-flammatory, profibrotic signaling pathways are driven by proximal tubular cells with a failed repair transcriptomic signature, proinflammatory fibroblasts and collecting duct cells. We identify GPRC5A as a marker for cyst-lining collecting duct cells that exhibits increased transcription factor binding motif availability for NF-κB, TEAD, CREB and retinoic acid receptors. We identify and validate a distal enhancer regulating GPRC5A expression containing these motifs. This single cell multiomic analysis of human ADPKD reveals previously unrecognized cellular heterogeneity and provides a foundation to develop better diagnostic and therapeutic approaches.

Autosomal dominant polycystic kidney disease (ADPKD) affects ~1 in 400–1000 individuals worldwide and is the most common inherited cystic disease[1]. Cyst growth and expansion in ADPKD ultimately destroys normal kidney tissue leading to chronic kidney disease (CKD) and often progresses to end stage kidney disease (ESKD). Most cases are characterized by mutations in either the *PKD1* or *PKD2* genes that encode polycystin-1 (PC-1) and polycystin-2 (PC-2), respectively. PC-1 and PC-2 are transmembrane proteins that form a heterodimeric complex localized to the primary cilium, plasma membrane and endoplasmic reticulum. Some evidence suggests that PC-1 and PC-2 sense fluid flow and regulate intracellular calcium levels[2], but this

remains controversial and the primary function of these proteins is undefined. Other pathways that have been implicated in PC-1 and PC-2 signaling include cAMP, mammalian target of rapamycin complex (mTORC), WNT, metabolic pathways including glycolysis and mito-chondrial function[1,2]. The vasopressin receptor antagonist tolvaptan has been approved to slow the progression of ADPKD and acts by decreasing cAMP concentration, although therapy is associated with polyuria which can limit tolerance[3]. Accordingly, the development of new therapeutic approaches to ADPKD is of paramount importance.

Recent advances in single-cell or single-nucleus RNA sequencing (scRNA-seq or snRNA-seq) technologies have advanced our

[1]Division of Nephrology, Department of Medicine, Washington University in St. Louis, St. Louis, MO, USA. [2]Department of Pathology and Immunology, Washington University in St. Louis, St. Louis, MO, USA. [3]Chinook Therapeutics, Inc., Seattle, WA, USA. [4]Chinook Therapeutics, Inc., Vancouver, BC, Canada. [5]Department of Medicine, University of Maryland School of Medicine, Baltimore, MD, USA. [6]Department of Physiology, University of Maryland School of Medicine, Baltimore, MD, USA. [7]Johns Hopkins School of Medicine, Baltimore, MD, USA. [8]Department of Developmental Biology, Washington University in St. Louis, St. Louis, MO, USA. ✉e-mail: humphreysbd@wustl.edu

understanding of cell types and states present in both healthy and diseased human kidney[4–6]. snRNA-seq in particular is well-suited for the analysis of human tissue since it is compatible with cryopreserved samples and we have demonstrated comparable sensitivity to scRNA-seq[4]. Recent single-cell profiling approaches have been extended to include the epigenome. The single-nucleus assay for transposase-accessible chromatin sequencing (snATAC-seq) technique utilizes hyperactive Tn5 transposase to map accessible chromatin at single-cell resolution[7,8]. The resulting large datasets can be used to predict *cis*-regulatory DNA networks and transcription factor activity[9,10], providing complementary information to snRNA-seq[11]. We and others have recently reported multimodal single-cell atlases of human and mouse kidneys and leveraged these atlases to redefine cellular heterogeneity, demonstrating the potential utility of multimodal single-cell analyses to refine our understanding of kidney biology[12–14]. Furthermore, recent advances in epigenetic editing technology with dCas9 fusion protein enable us to validate gene regulatory networks predicted by snATAC-seq analysis[15].

Here, we have performed snRNA-seq and snATAC-seq on eight human ADPKD kidneys to understand the cell states and dynamics in late stage ADPKD at single-cell resolution. We successfully identified previously unrecognized subpopulations and their molecular signatures in cyst-lining cells and other cell types. Unexpectedly, most proximal tubular cells in ADPKD kidneys, whether cystic or not, had adopted a profibrotic failed-repair transcriptomic signature. We could also identify proinflammatory fibroblast and collecting duct subtypes with evidence of activation of inflammatory pathways in ADPKD kidneys. We observed specific upregulation of the orphan G protein-coupled receptor GPRC5A in collecting duct cysts, and we identified a distal *GPRC5A* enhancer regulating its expression in ADPKD principal cells. We generated an interactive data visualization tool encompassing both transcriptomic and epigenomic data (http://humphreyslab.com/SingleCell/). Our study represents the first multimodal single-cell atlas of human ADPKD and reveals new cell states associated with late stage disease.

## Results

### Single-cell transcriptional and chromatin accessibility profiling on ADPKD kidneys

We performed snRNA-seq on eight ADPKD and five control adult kidney samples with 10X Genomics Chromium Single Cell 3′ v3 chemistry (Fig. 1a). The ADPKD patients ranged in age from 35 to 61 years and included men ($n = 4$) and women ($n = 4$). All patients had severely impaired renal function requiring kidney transplantation at the time of sample collection (mean eGFR = 13.6 ml/min/1.73 m$^2$); 1 patient had been on maintenance dialysis for 6 months prior to transplant and the other patients ($n = 7$) received pre-emptive transplantation (Supplementary Table 1). The genetic cause of ADPKD was not known in these patients. The ADPKD samples were collected from the base (cup) of large superficial cysts (Supplementary Fig. 1).

Five control kidneys samples were obtained from the outer cortex of non-tumor kidney tissue nephrectomized from patients with preserved renal function (mean eGFR = 72.8 ml/min/1.73 m$^2$) that ranged in age from 50 to 62 years and included men ($n = 3$) and women ($n = 2$). These five samples have been reported previously using the 10X Chromium 5′ Chemistry[12]; however, we generated new libraries from these samples using the 3′ chemistry in order to allow direct comparison with the ADPKD samples. After batch quality control (QC) filtering and preprocessing, ADPKD or control snRNA-seq datasets were integrated with Seurat[16] and visualized in UMAP space to annotate cell clusters (Supplementary Figs. 2a and 3a, See also Methods). Interestingly, lineage marker expression was largely preserved in ADPKD kidneys (Supplementary Fig. 3a), allowing the assignment of cell of origin for cyst cells. All major tubular cell types were identified in both ADPKD and control datasets (Supplementary Figs. 2a and 3a) but

leukocyte cluster was not detected in control dataset (Supplementary Fig. 2a). After removing doublets and low-quality clusters, we obtained a total of 102,710 nuclei by snRNA-seq; 62,073 nuclei from ADPKD and 40,637 nuclei from control kidneys (Supplementary Figs. 2b and 3b). Control kidney datasets had more unique genes and transcripts per cell than ADPKD samples (Supplementary Fig. 4).

We dissociated ADPKD tissues into nuclear suspensions using the exact same protocol as we did on control kidneys, but we observed more debris in nuclei suspension from ADPKD kidneys likely reflecting the fibrotic nature of late stage CKD samples (Supplementary Fig. 4, See also Methods). ADPKD and control kidney datasets were integrated in Seurat, and batch correction was performed with the R package Harmony[17] (Fig. 1a, b). Differentially expressed genes were identified (Supplementary Data 1–3, Fig. 1c), and cell types were assigned to each of the unsupervised clusters based on lineage markers (Fig. 1b, c, Supplementary Table 2). The two smallest clusters expressed uroepithelial marker genes (*UPK3A*, *PSCA*), suggesting that they were probably uroepithelium (Fig. 1b). Nearly all of these cells (~99%) were from one patient (PKD8). We observed considerable variability in cell-type frequency in ADPKD samples compared to controls (Supplementary Fig. 5). This variability may reflect the location of the cyst that was sampled (cortical, corticomedullary or medullary region). Most ADPKD cases arise from mutations in either the *PKD1* or *PKD2* gene, although the genetic cause of ADPKD was not known in these patients. We compared *PKD1* and *PKD2* mRNA expression between healthy and ADPKD samples. Overall expression was very low, but expression of these genes was higher in ADPKD (Supplementary Fig. 6).

snATAC-seq (10X Genomics Chromium Single Cell ATAC v1) was also performed on the same ADPKD samples to profile single-cell chromatin accessibility. snATAC-seq datasets on control kidneys were previously described[12]. Multiomic integration and label transfer with Seurat was performed on ADPKD and control kidney datasets[11] (Fig. 2a, See also Methods). The prediction scores for label transfer in the healthy kidneys is higher than that of the ADPKD dataset, likely reflecting the lower gene detection per cell in the ADPKD samples (Supplementary Fig. 7a, b). The snATAC-seq datasets were filtered using an 80% confidence threshold for cell-type assignment to remove heterotypic doublets, and we obtained 33,621 nuclei for control and 17,365 nuclei for ADPKD. Finally, ADPKD and control snATAC-seq datasets were integrated with Harmony (Fig. 2a) and visualized in UMAP space (Fig. 2b). Differentially accessible regions (DAR) among the clusters include the genomic regions around the transcription start site (TSS) of kidney lineage marker genes (Fig. 2c). Each cluster was annotated based on gene activities (Fig. 2d). We confirmed that snATAC-seq cell-type predictions obtained by label transfer (Supplementary Fig. 7c, d) and curated annotations of unsupervised clusters based on gene activities (Fig. 2b–d, Supplementary Table 3) were largely consistent. We performed downstream analyses with these cell-type assignments based on unsupervised clustering and gene activities of lineage markers (Fig. 2b–d, Supplementary Data 4–12). The number of DAR in each cell type of ADPKD was less than that of control, suggesting generally less chromatin accessibility in ADPKD kidney cells (Supplementary Fig. 8). Given the lower quality of the snRNA-seq libraries from ADPKD samples (Supplementary Fig. 4), we hypothesize that the systematically lower DAR in the ADPKD samples may also reflect reduced chromatin quality.

### Activation of inflammatory, profibrotic pathways in ADPKD kidneys

To interrogate cellular responses in ADPKD we performed geneset enrichment analysis with hallmark genesets using the Molecular Signatures Database (MsigDB) on the whole snRNA-seq dataset (Fig. 3a) using Vision, a tool for annotating sources of variation in single-cell RNA-seq data[18]. We observed that various inflammatory pathways (IL6-

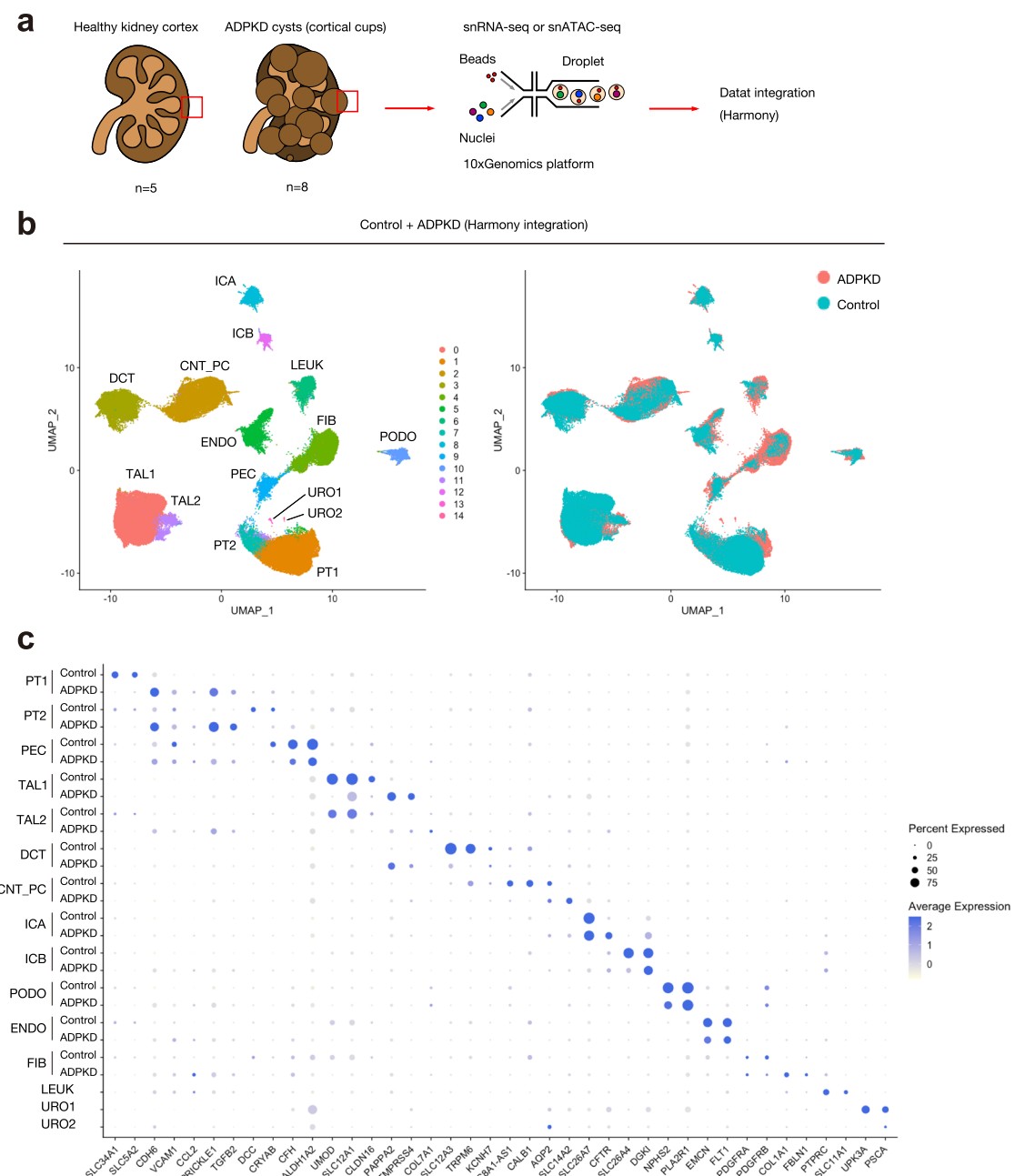

**Fig. 1 | Single-nucleus transcriptional profiling on human ADPKD kidneys.**
**a** Overview of experimental methodology. $n = 8$ human ADPKD kidneys and $n = 5$ control kidneys were analyzed with snRNA-seq and snATAC-seq. Batch effect on the integrated datasets was corrected with Harmony. See Method section for detail. **b** UMAP plot of integrated snRNA-seq dataset with annotation by cell type (left) or disease condition (right). PT proximal tubule, PEC parietal epithelial cells, TAL thick ascending limb of Henle's loop, DCT distal convoluted tubule, CNT_PC connecting tubule and principal cells, ICA Type A intercalated cells, ICB Type B intercalated cells, PODO podocytes, ENDO endothelial cells, FIB fibroblasts, LEUK leukocytes, URO uroepithelium. **c** Dot plot of snRNA-seq dataset showing gene expression patterns of cluster-enriched markers for ADPKD or control kidneys. For LEUK and URO1/2 clusters, data from ADPKD kidneys were shown. The diameter of the dot corresponds to the proportion of cells expressing the indicated gene and the intensity of the dot corresponds to average expression relative to all cell types.

mediated STAT3 activation and NF-κB activation) as well as TGFβ signaling pathway were generally activated in the ADPKD microenvironment (Fig. 3a, b). Inflammation has been previously shown to be associated with progression of ADPKD[19]. TGFβ signaling drives profibrotic pathways in chronic kidney disease, including ADPKD. Next, we performed motif enrichment analysis of the transcription factors in the whole-snATAC-seq dataset. In agreement with the geneset enrichment analysis, we found the motifs of transcription factors related to NF-κB pathway (RELA), IL6-induced inflammation (STAT3) and TGFβ signaling (SMAD2::SMAD3::SMAD4) were enriched in many cell types of ADPKD kidneys. To infer the source of these signals, we performed

ligand-receptor analysis with CellChat[20] to quantitatively infer cell–cell communication networks. We identified three primary cell types and the ligand they predominately secreted (Fig. 4a, b): IL6 by fibroblasts, TNFα by collecting duct epithelial cells and TGFβ by proximal tubular cells (PTC).

**Proximal tubular cells exhibit a failed-repair cell state in ADPKD kidneys**
We observed PTC expressing *VCAM1* in both ADPKD and control kidney datasets (Fig. 1c, Supplementary Figs. 2b and 3b). *Vcam1* + PTC were recently described as failed-repair proximal tubular cells

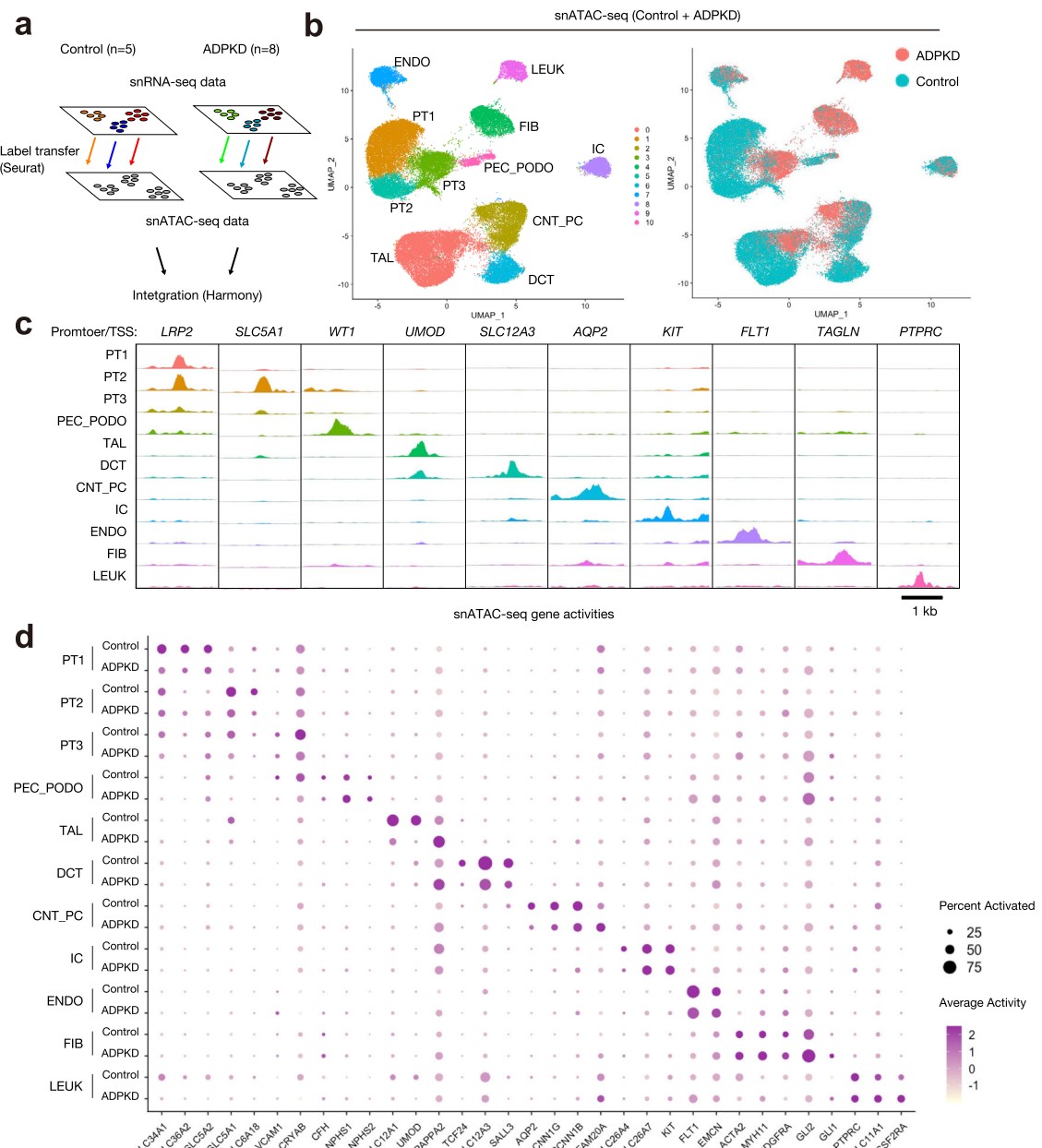

**Fig. 2 | Single-nucleus chromatin accessibility profiling on human ADPKD kidneys. a** Graphical abstract of multimodal integration strategy for the snATAC-seq datasets. The integrated ADPKD or control snATAC-seq datasets were label-transferred from cognate snRNA-seq datasets, and the snATAC-seq datasets were filtered using an 80% prediction score threshold for cell-type assignment. After filtering, control and ADPKD datasets were merged, and batch effect was corrected with Harmony. See Supplementary Fig. 7 and Method section for detail. **b** UMAP plot of snATAC-seq dataset with gene activity-based cell-type assignments (left) or annotation by disease condition (right). **c** Fragment coverage (frequency of Tn5 insertion) around the differentially accessible regions (DAR) around each cell type at lineage marker gene transcription start sites. Scale bar indicates 1 Kb. **d** Dot plot of snATAC-seq dataset showing gene activity patterns of cluster-enriched markers for control or ADPKD kidneys. The diameter of the dot corresponds to the proportion of cells with detected activity of indicated gene and the intensity of the dot corresponds to average gene activity relative to all cell types.

(FR-PTC) with a proinflammatory and profibrotic transcriptional profile in mouse kidneys[21–23]. Furthermore, *VCAM1* + PT in human kidneys were shown to have close gene expression signatures to FR-PTC in mice, suggesting that human *VCAM1* + PT closely resemble mouse FR-PTC[12]. Subclustering of the PT lineage (PT-1 and PT-2, Fig. 1b) in control kidneys identified FR-PTC as *VCAM1* + PT subpopulation (Fig. 5a). To compare the FR-PTC in control kidneys and PT cells in ADPKD kidneys, we performed subclustering analysis on all ADPKD PT cells with FR-PTC alone from controls. This allowed us to identify 4 subclusters (PT-1/2/3/4), Fig. 5b, c). The FR-PTC originating from control PT were clustered into PT-1/2 (47.1% and 49.4% of control cells, respectively). In

contrast, almost all (>99%) PT-3/PT-4 were derived from ADPKD PT cells.

Next, we compared the gene expression of these PT subtypes in ADPKD kidneys and those in mouse ischemia reperfusion injury (IRI) datasets (Fig. 5d). The expression of mouse FR-PTC was better correlated with human FR-PTC in control kidneys or ADPKD PT subtypes compared to normal PT in control kidneys. The correlation of gene expression between mouse FR-PTC and PT-3 in ADPKD was weaker than other ADPKD subtypes, although the gene signature of PT-3 was still closer to mouse FR-PTC than to other mouse PT subtypes. Together, the gene expression of ADPKD PT subtypes generally correlated well with mouse FR-PTC from IRI datasets, suggesting that most of the

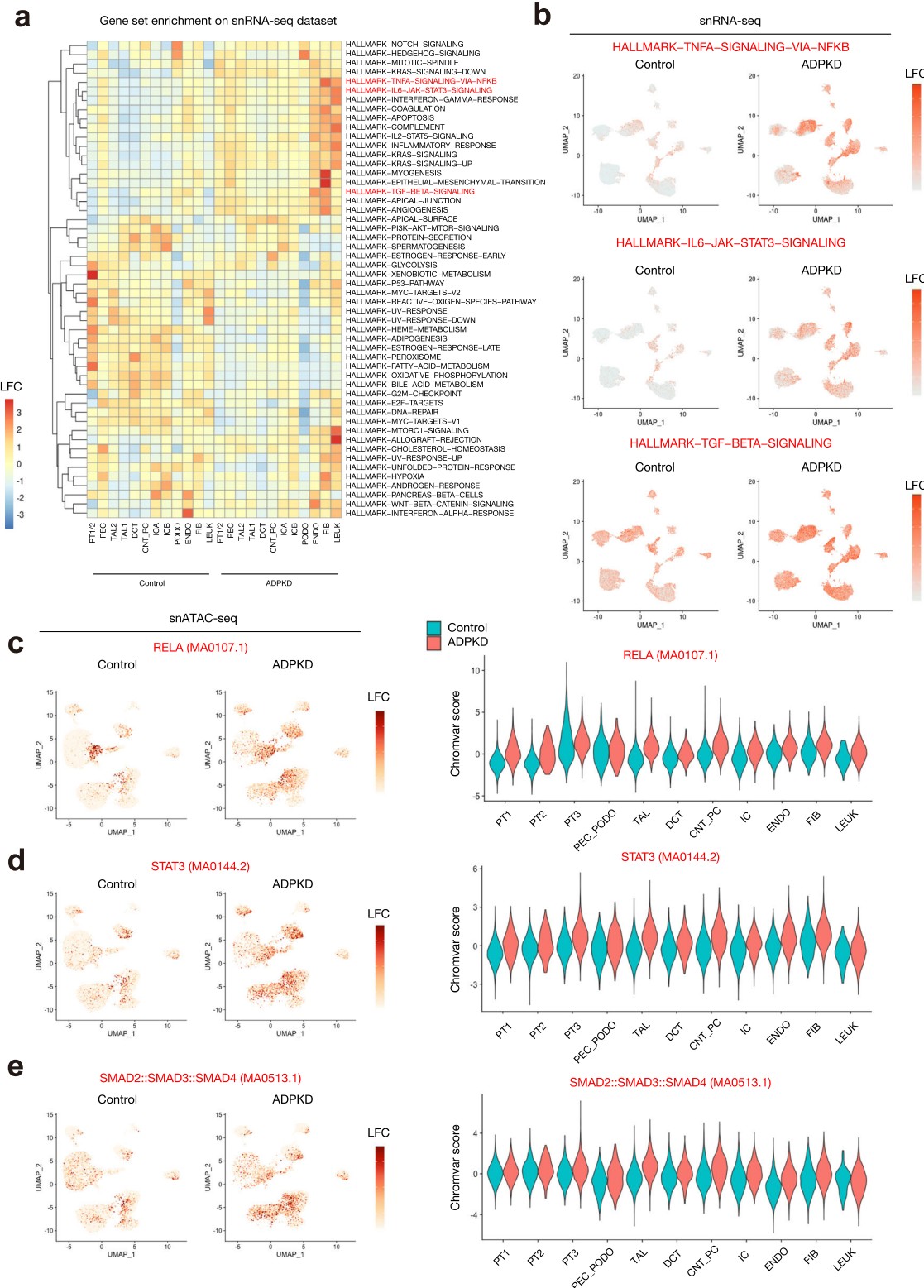

**Fig. 3 | Activation of proinflammatory, profibrotic pathways in ADPKD kidneys. a** Heatmap showing enrichment of hallmark genesets of the Molecular Signatures Database (MsigDB) in each cell type of ADPKD or control kidneys. Source data are provided as a Source Data file. **b** UMAP displaying enrichment of genes regulated by NF-κB pathway in response to TNFα (upper), genes upregulated by IL6 via STAT3 (middle) or genes upregulated in response to TGFβ signaling (lower) in snRNA-seq dataset. **c–e** UMAP displaying enrichment of transcription factor binding motifs in control or ADPKD kidneys (left) and violin plot showing the relative motif enrichment scores in each cell type (right) for RELA (**c**), STAT3 (**d**), or SMAD2/SMAD3/SMAD4 complex (**e**). The color scale represents a normalized log-fold-change (LFC).

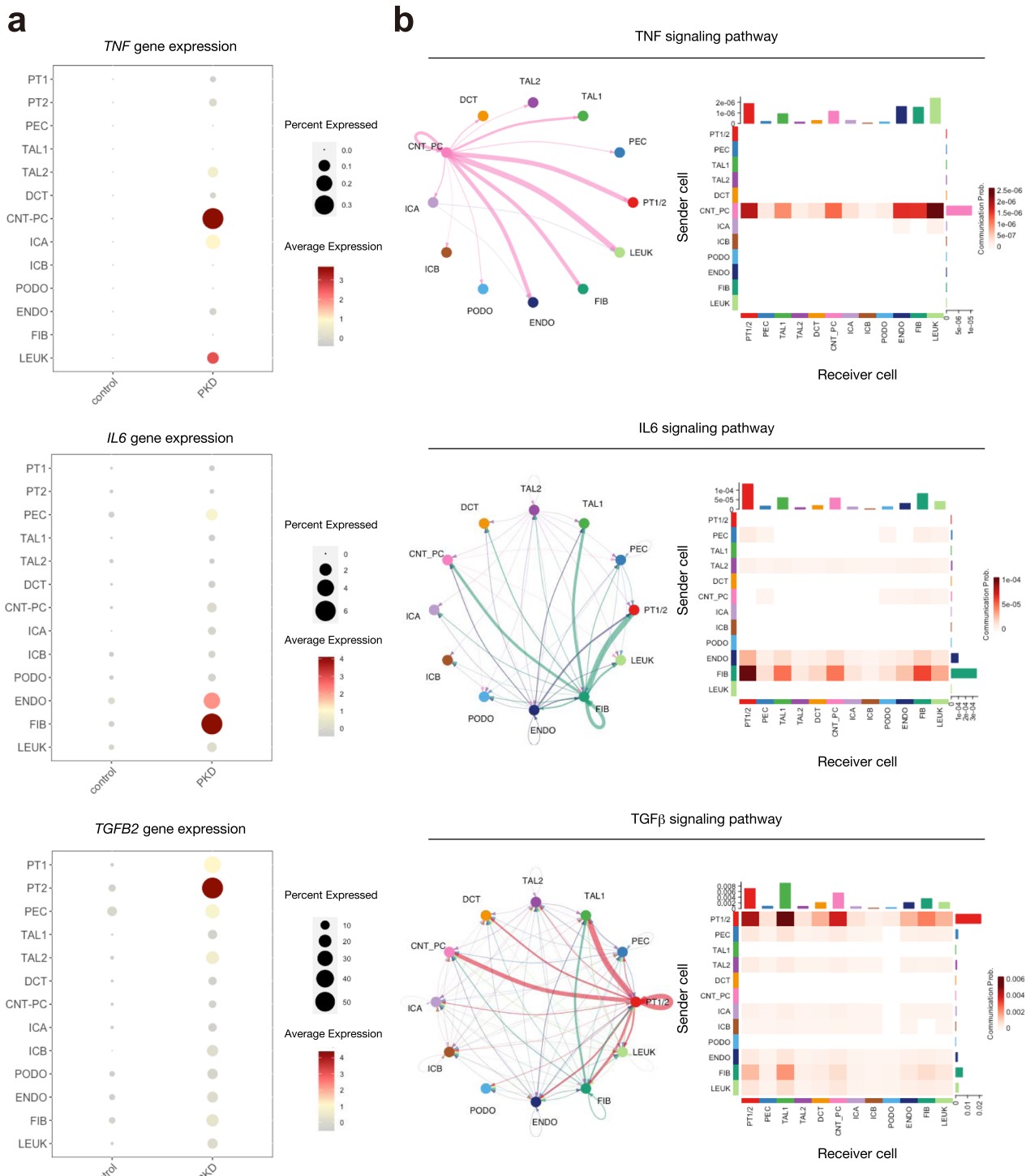

**Fig. 4 | Ligand-receptor analysis identified proinflammatory, profibrotic signaling network. a** Dot plot showing gene expression of *TNF* (upper), *IL6* (middle), or *TGFB2* (lower) in each cell type in ADPKD or control kidneys. The diameter of the dot corresponds to the proportion of cells expressing the indicated gene and the intensity of the dot corresponds to average expression relative to all cell types. **b** Ligand-receptor analysis with CellChat. Circle plot showing an inferred network (left) or heatmap (right) showing communication probabilities from senders (secretors) to receivers (targets) for TNF signaling pathway (upper), IL6 signaling pathway (middle), or TGFβ signaling pathway (lower). Thickness of an arrow in a circle plot indicates interaction strength.

PT cells—even non-cystic ones—in ADPKD kidneys adopt a FR-PTC transcriptional signature. We cannot rule out the possibility that the strong FR-PTC signature in ADPKD PT might reflect some degree of dissociation bias through, for example through preferential loss of healthy PT cells during dissociation.

While PT-3/4 showed lower *VCAM1* expression compared to PT-1, they also expressed higher levels of failed-repair signature genes

(*CREB5, TPM1, PROM1*[*CD133*]*, TGFB2*, and *HAVCR1*, Fig. 5c)[12,21], suggesting that *VCAM1* may not be a sole defining marker of the FR-PTC state. We also observed various levels of PT marker gene expression among PT subtypes in ADPKD (Fig. 5e). *LRP2* was expressed among all the subtypes while *CUBN* was mainly expressed in PT-1/PT-2 (Fig. 5e). Co-immunostaining of VCAM1 with CUBN in ADPKD samples (Fig. 5f) suggests heterogeneity of CUBN expression among VCAM1+ non-

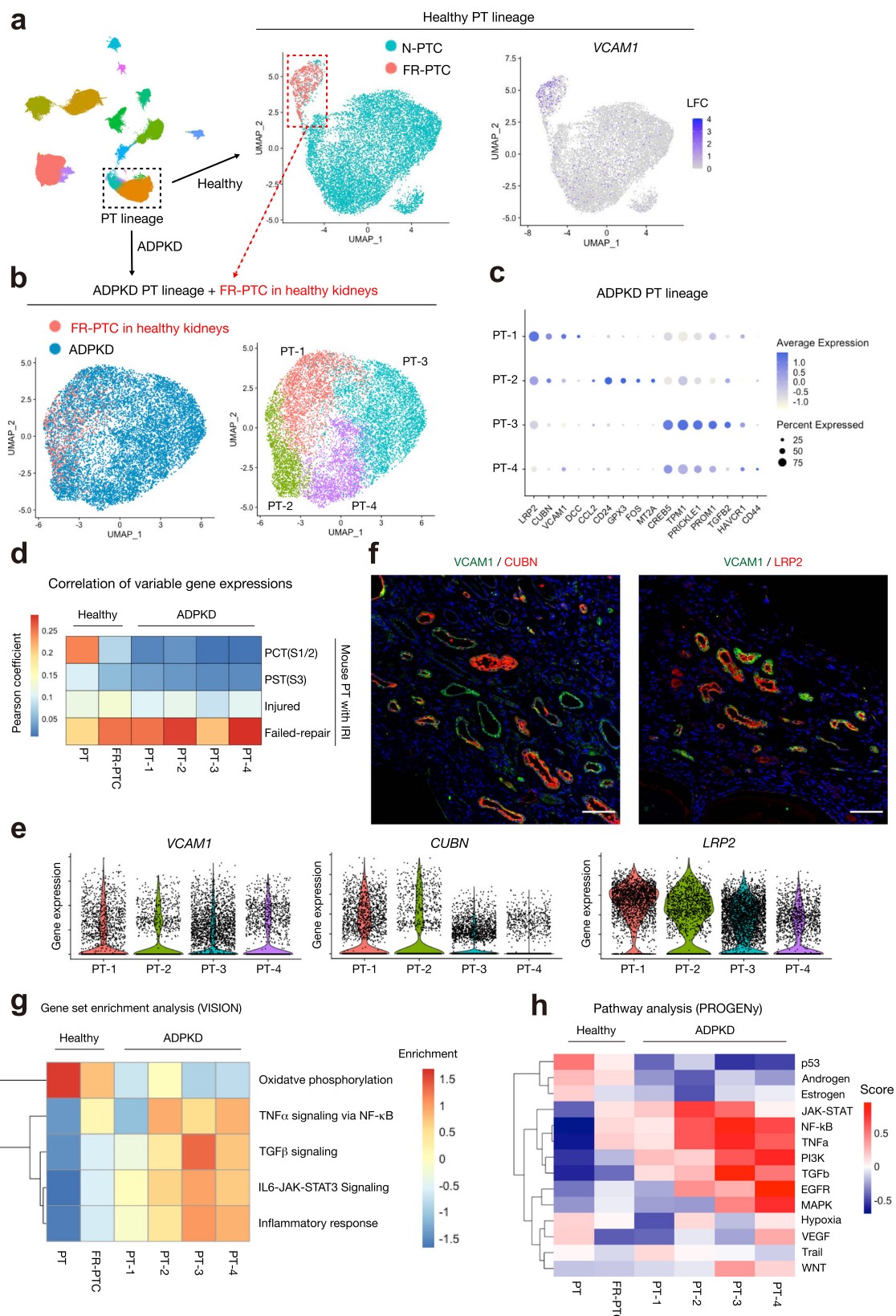

cystic tubular cells. By contrast, nearly all VCAM1 + cells were also LRP2 + , suggesting that VCAM1 + cells were of proximal tubular cell lineage. These VCAM1 + , non-cystic tubules did not necessarily appear atrophic, suggesting that non-cystic PT in ADPKD kidneys adopt a stressed FR-PTC state, perhaps the result of chronic hypoxia in fibrotic interstitium.

Next, we analyzed the PT subcluster data with geneset enrichment analysis. FR-PTC in control and PT subtypes in ADPKD kidneys lost oxidative phosphorylation gene signatures and gained inflammatory signatures (Fig. 5g), consistent with the previously reported mitochondrial dysfunction in human kidneys with CKD and inflammatory responses caused by loss of mitochondrial integrity[24]. Of note, each of

**Fig. 5 | Proximal tubular cells express a failed-repair molecular signature in ADPKD kidneys. a** Subclustering of healthy control PT lineage on the UMAP plot of snRNA-seq dataset, colored by subtypes (left and middle) or *VCAM1* expression level (right). N-PTC normal proximal tubular cells, FR-PTC failed-repair proximal tubular cells. **b** Subclustering of healthy control FR-PTC and ADPKD PT cells on the UMAP, colored by disease (left) or subtypes (PT-1/2/3/4, right). **c** Dot plot showing gene expression patterns of the genes enriched in each of PT subtypes in ADPKD kidneys. The diameter of the dot corresponds to the proportion of cells expressing the indicated gene and the intensity of the dot corresponds to average expression relative to all ADPKD PT cells. **d** Pearson correlations of the averaged expressions of highly variable genes between PT subtypes in ischemia reperfusion injury (IRI)

model mouse kidneys (GSE139107) and those of human ADPKD dataset. The highly variable genes among IRI mouse PT cells that also exist in human dataset were analyzed (1648 genes). The heatmap shows Pearson correlation coefficients (*R*). Source data are provided as a Source Data file. **e** Violin plot showing *VCAM1*, *CUBN*, or *LRP2* gene mRNA expression among PT subtypes of ADPKD kidneys. **f** Immunohistochemistry analysis on human ADPKD kidney for VCAM1 (green) and CUBN (red, left) or LRP2 (red, right). Representative images of *n* = 3 samples. Scale bar indicates 50 μm. **g** Heatmap showing enrichment of hallmark genesets of the Molecular Signatures Database (MsigDB) for oxidative phosphorylation or inflammatory pathways. Source data are provided as a Source Data file. **h** Heatmap showing pathway enrichment on PT subpopulations with PROGENy.

the ADPKD PT subtypes showed differential enrichment of inflammatory genesets, suggesting heterogeneity of inflammatory pathways activated in ADPKD PT cells. These findings were further supported by PROGENy pathway analyses[25] (Fig. 5h). These analyses also suggest TGFβ pathway activation in ADPKD cells, especially in PT-3/4. *TGFB2* was differentially expressed among ADPKD PT subtypes (Supplementary Fig. 9, Supplementary Data 13). *TGFB2* was also predicted to play a dominant role in TGFβ signaling in ligand-receptor analysis with Cell-Chat (Supplementary Fig. 10). Interestingly, the ligand-receptor analysis suggests PTC themselves are also the major target of TGFβ signaling (Fig. 4b). In agreement with these findings, the expression of TGFβ receptor *TGFBR2* was upregulated in ADPKD PT cells, especially in PT-3/4 (Supplementary Fig. 11a). Furthermore, the binding motif of SMADs that are downstream effectors of TGFβ signaling was enriched in ADPKD PT in the snATAC-seq dataset (Supplementary Fig. 11b). These findings suggest that TGFβ secreted from PT may be acting in an autocrine or paracrine fashion in addition to signaling to other surrounding cells in ADPKD. Collectively, these findings indicate that the increased and heterogeneous FR-PTC in ADPKD kidneys adopt a proinflammatory, profibrotic cell state.

## Expansion of proinflammatory fibroblast subtypes in ADPKD kidneys

ADPKD progression is associated with CKD and interstitial fibrosis[26]. IL6 was predicted to be mainly secreted by the fibroblasts in ADPKD kidneys (Fig. 4a), suggesting a proinflammatory role of fibroblasts in ADPKD microenvironment. To characterize the alteration in molecular signatures of interstitial cells in ADPKD kidneys, we performed subclustering on fibroblast (FIB) clusters, resulting in separation of 7 subclusters (Fig. 6a, b). Two uncharacterized clusters (Unknown1 and Unknown2), expressed tubular cell markers (*SLC12A1*, *SLC34A1*, and *LRP2*) that were detected in both ADPKD and control kidneys, most likely residual doublets despite our use of DoubletFinder[27]. FIB1 and FIB2 expressed *PDGFRB*, and they were detected in both ADPKD and control kidneys. In contrast, *ACTA2*+ myofibroblasts (MyoFIB) were exclusively detected in ADPKD kidneys. There was also another ADPKD-specific cluster (PKD-FIB) with a distinct molecular signature (Fig. 6b, Supplementary Data 14). PKD-FIB expressed *IL6* and *FGF14* at high levels (Fig. 6b), suggesting this subset as the major source of IL6 in ADPKD kidneys. Indeed, the ligand-receptor analysis among FIB subtypes along with all the other cell types (Supplementary Fig. 12a) suggested that most of IL6 signaling was from PKD-FIB, and the major target cells were PT in ADPKD kidneys. This finding was consistent with JAK-STAT pathway activation (Fig. 5g, h) as well as STAT3 motif enrichment in ADPKD PT (Fig. 3c). Each of these subtypes had variable expression levels of fibroblast marker genes (Figs. 1c and 6c), suggesting cellular heterogeneity of fibroblasts. Pathway analysis with PROGENy[25] indicated TGFβ signaling pathway activation in the MyoFIB cluster, as well as TNFα-induced NF-κB activation in the PKD-FIB cluster (Fig. 6d). Since *TGFB2* expression was upregulated in ADPKD PT (Figs. 4c and 5c), cyst FR-PTC may be driving interstitial myofibroblast proliferation. *TNF* expression was mainly detected in CNT_PC cluster of ADPKD kidneys, and ligand-receptor analysis suggested that the

CNT_PC is the major source of TNF in ADPKD kidneys (Fig. 4a, b). Ligand-receptor analysis on all cell types along with FIB subtypes suggested PKD-FIB as a target of TNFα signaling, although other targets include PT, ENDO, and LEUK clusters (Supplementary Fig. 12b).

We performed deconvolution analysis on published microarray data of human ADPKD kidneys[28] with our dataset using CIBERSORTx, a machine learning method that imputes gene expression profiles and estimates the frequency of cell types in a mixed cell population[29]. This deconvolution analysis predicted a significant increase in the fibroblasts (FIB) population in cystic kidneys compared to either minimally cystic or normal kidneys (Fig. 6e). Cell-type-specific expression purification at high resolution with CIBERSORTx indicated upregulation of Myo-FIB markers (*ACTA2* and *FN1*) and PKD-FIB markers (*IL6* and *FGF14*), suggesting expansion of these FIB subsets (Fig. 6f). These results confirm that PKD-FIB and Myo-FIB subsets are associated with large cysts in ADPKD kidneys.

## Characterization of collecting duct cyst subtypes

In ADPKD kidneys, cysts are derived from both proximal and distal nephron segments. Most cyst-lining cells expressed the distal nephron marker CDH1 in our samples. To characterize the cyst-lining cells that originated from principal cells, we performed reclustering of the connecting tubule and principal cell (CNT_PC) cluster (Fig. 7a, Supplementary Data 15), resulting in separation of 8 clusters. Among them, two clusters exhibited enriched expression of *SLC26A7* (IC-A marker) or *LRP2* (PT marker) transcripts along with mitochondrial genes, suggesting that they were multiplets including intercalated (IC) or PT cells. Cluster1 and 2 were detected in both healthy and ADPKD kidney cells (Fig. 7a, Supplementary Fig. 13). Based on the differentially expressed genes, we annotated these as normal connecting tubules (N-CNT, cluster1) and normal PC (N-PC, cluster2), respectively. We also detected four ADPKD-specific clusters (PKD-CNT [ADPKD-specific connecting tubular cells] and PKD-CDC1, 2 and 3 [ADPKD-specific collecting duct cells]). PKD-CNT (cluster5) and PKD-CDC1 (cluster0, Supplementary Fig. 13) differentially expressed *MET*, and PKD-CDC2 (cluster4) differentially expressed *LCN2*. *MET* and *LCN2* were previously found to be essential for disease progression in an ADPKD mouse model[30,31] (Fig. 7b). PKD-CDC3 (cluster7) represented a smaller cyst subcluster with unique markers (Fig. 7b). Interestingly, ADPKD subclusters upregulated the expression of *CFTR* (Fig. 7b and Supplementary Fig. 14), which encodes a chloride ion channel thought to be involved in cyst fluid accumulation[32].

A majority of TNFα signaling was predicted to be from CNT_PC cluster in our ligand-receptor analysis (Fig. 4), and TNFα was primarily detected in the PKD-CDC2 subset (Fig. 7c), suggesting that PKD-CDC2 is a major source of TNFα signaling in ADPKD kidneys. To characterize the molecular signatures of PKD-CDC2 as well as other PKD-specific subtypes, we applied geneset enrichment analyses on CNT_PC subclusters (Fig. 7d). This revealed general down-regulation of genes related to oxidative phosphorylation in the ADPKD clusters, and enrichment of hypoxic response and glycolysis gene signatures in PKD-CDC2 (Fig. 7e) as well as inflammatory gene expression signatures (Fig. 7e). These findings are consistent with the previously observed

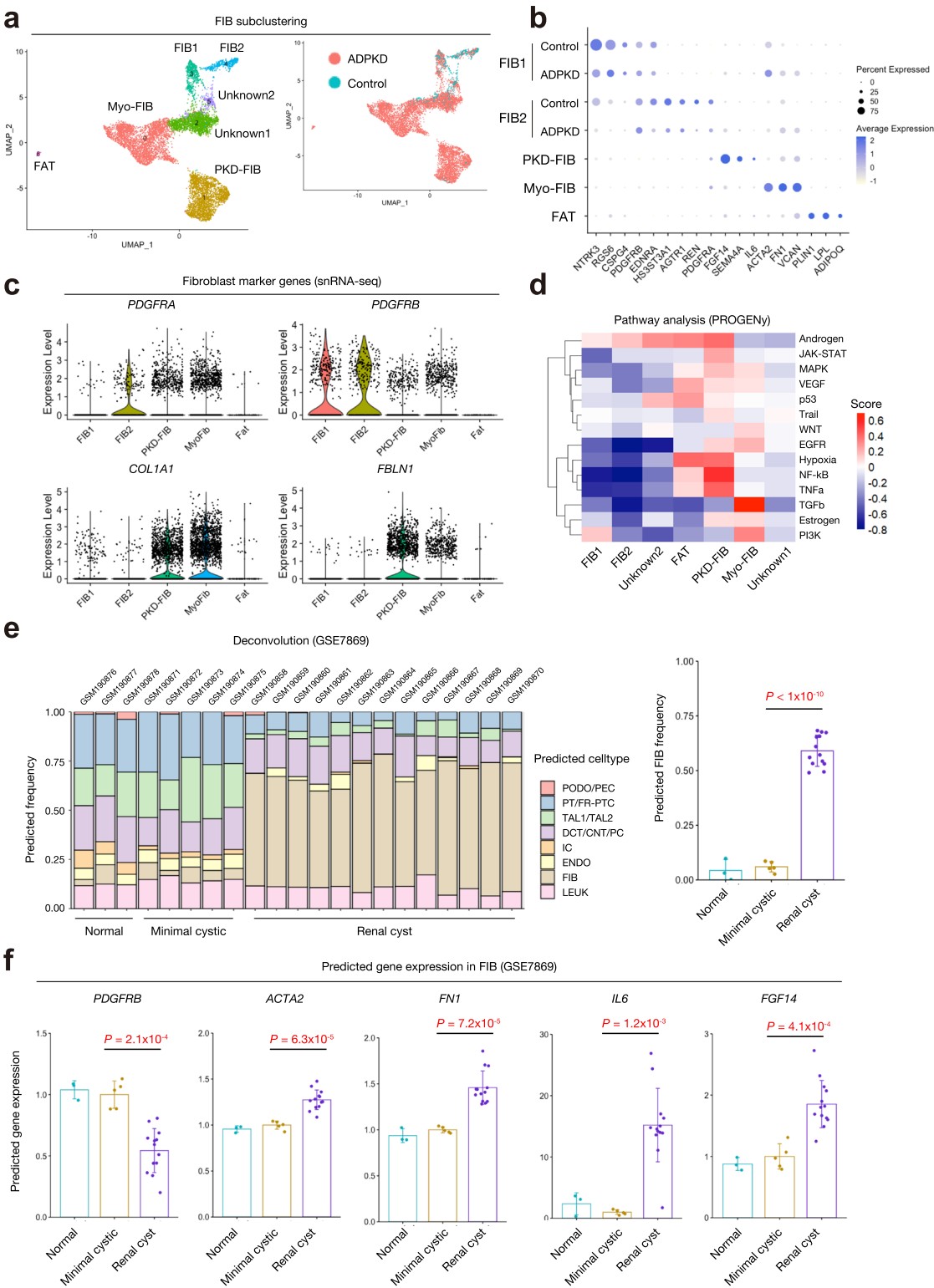

response to hypoxia, activation of glycolysis and inflammation in ADPKD cysts[19,32].

Among the differentially expressed genes among CNT_PC subtypes, we identified *GPRC5A* as one of the most differentially upregulated genes in PKD-CDC1 cluster ($P\ adj. < 1.0 \times 10^{-300}$, Log-fold change = 1.62) as well as PKD-CDC2 cluster ($P\ adj. < 2.0 \times 10^{-90}$, Log-fold change = 1.01, Fig. 7f, Supplementary Data 15). Immunofluorescence studies validated GPRC5A expression in CDH1 + cyst-lining cells (Fig. 7g, left), while its expression was faint in CDH1 + tubular cells in control kidneys (Fig. 7g, right). GPRC5A expression was recently found to be

upregulated by hypoxia, and to suppress the Hippo pathway, promoting cell survival in vitro[33]. Thus, GPRC5A might confer resistance to hypoxia in cyst cells, potentially promoting cyst growth in ADPKD.

Another interesting gene upregulated in PKD-CDC1 ($P\ adj. = 2.7 \times 10^{-14}$, Log-fold change = 0.30) and PKD-CDC2 ($P\ adj. = 8.0 \times 10^{-234}$, Log-fold change = 1.19, Supplementary Data 15) was *MIR31HG* that encodes a long non-coding RNA (Supplementary Fig. 15a). *MIR31HG* expression was found to be induced by hypoxia to promote HIF1A-dependent gene expression as a co-activator[34], consistent with enrichment of glycolysis-related genes (Fig. 7e). Furthermore,

**Fig. 6 | Expansion of proinflammatory, profibrotic fibroblast subtypes in ADPKD kidneys. a** Subclustering of FIB on the UMAP plot of snRNA-seq dataset with annotation by subtype (left) or disease condition (right). MyoFIB Myofibroblast, PKD-FIB ADPKD-specific fibroblast subtype, FAT adipocytes. **b** Dot plot showing gene expression patterns of the genes enriched in each of FIB subpopulations. For FIB1 and FIB2, control and ADPKD data were individually shown. The diameter of the dot corresponds to the proportion of cells expressing the indicated gene and the intensity of the dot corresponds to average expression relative to all FIB cells. **c** Violin plot showing fibroblast marker gene expression among FIB subclusters; *PDGFRA* (upper left), *PDGFRB* (upper right), *COL1A1* (lower left), and *FBLN1* (lower right). **d** Heatmap showing pathway enrichment on FIB subpopulations with PROGENy. The color scale represents pathway enrichment score. **e** Predicted frequencies of cell types in each dataset of normal kidney cortex (*n* = 3) of healthy control, and minimal cystic tissue (*n* = 5) or renal cyst (*n* = 13) of ADPKD patients in deconvolution analysis of human ADPKD kidney datasets (GSE7869) with CIBERSORTx. The predicted FIB frequencies in each group are also shown (right). Source data are provided as a Source Data file. **f** Predicted relative gene expressions of *PDGFRB*, *ACTA2*, *FN1*, *IL6*, or *FGF14* in FIB of each group with CIBERSORTx. Each dot represents a biological replicate for normal kidney cortex (*n* = 3) of healthy control, and minimal cystic tissue (*n* = 5) or renal cyst (*n* = 13) of ADPKD patients. Bar graphs represent the mean and error bars are the s.d. One-way ANOVA with post hoc Dunnett's multiple comparisons test. Source data are provided as a Source Data file.

*MIR31HG* was shown to prevent cellular senescence via suppression of *CDKN2A* transcription through recruitment of polycomb group proteins[35]. Previously published studies concluded that senescence attenuates disease progression in a mouse model of ADPKD[36]. This suggests that *MIR31HG* may promote cyst growth through suppression of senescence and adaptation to hypoxia. While *CDKN1B* and *CDKN1C* expression was higher in normal subtypes (Supplementary Fig. 15b), senescence-related CDK inhibitors *CDKN1A* and *CDKN2A* were more abundant in PKD-CDC subtypes (Supplementary Fig. 15a, b), suggesting that cyst-lining cells may be prone to cellular senescence due to cellular stress. Upregulated *MIR31HG* expression in ADPKD kidneys (Supplementary Fig. 15a) could circumvent cellular senescence by inhibiting further upregulation of *CDKN2A*.

## Long-range genomic *cis*-interactions govern molecular signatures of cyst-lining cells

Next, we analyzed CNT_PC cluster in the snATAC-seq dataset to dissect the epigenetic mechanism driving unique molecular signatures in ADPKD cyst cells. We detected an ADPKD-specific PC subpopulation (PKD-CDC, Fig. 8a, Supplementary Data 16). This subtype showed higher gene activity of *GPRC5A* and *CD44* compared to other subpopulations (Fig. 8b), suggesting that they represent the combined PKD-CDC1 and PKD-CDC2 subclusters previously identified in snRNA-seq data. Transcription factor motif enrichment analysis indicated activation of NF-κB transcription factors and transcriptional enhanced associate domain (TEAD) family transcription factors in PKD-CDC (Fig. 8c, Supplementary Data 17). TEAD family transcription factors have been shown to play important roles in tissue homeostasis and organ size control, with activity regulated by their co-activator; YAP and TAZ in the Hippo pathway[37]. Recently, ROR1 was shown to be a co-receptor of Frizzled 1, activating YAP/TAZ by WNT5A/WNT5B[38]. Furthermore, ROR1 was found to mediate WNT5A-induced NF-κB activation[39,40]. Interestingly, we observed ROR1 protein expression in GPRC5A + cyst cells (Fig. 8d, Supplementary Fig. 16a). ROR1 expression in bulk tissue was directly correlated with cyst size in a reanalysis of published datasets[28] (Supplementary Fig. 16b). In agreement with this, *ROR1* expression was generally upregulated in ADPKD in our dataset (Supplementary Fig. 16c), although it was not in a cell-type-specific fashion (Supplementary Fig. 16d). Collectively, these findings suggest a potential role of ROR1 and GPRC5A in defining a unique gene signature of cyst-lining cells in ADPKD.

To characterize epigenetic mechanisms driving cyst growth, we predicted the *cis*-coaccessibility network in ADPKD kidneys around the *MIR31HG* or *GPRC5A* genes with Cicero[9] (Fig. 8e, Supplementary Fig. 15c). Although *MIR31HG* is differentially expressed in ADPKD cells (Fig. 7b), the promoter region of *MIR31HG* was more accessible in normal CNT cells (Supplementary Fig. 15c). However, we found that the 5′ distal region to *MIR31HG* gene was differentially accessible in ADPKD cells, and co-accessible to the *MIR31HG* promoter (Supplementary Fig. 15c). This 5′ distal region was previously shown to be an enhancer which upregulates *MIR31HG* expression during oncogenic cellular stress[35]. We performed CRISPR interference (CRISPRi) using

three sgRNAs targeting this 5′ distal region of *MIR31HG* in WT9-12 cells, an immortalized epithelial cell line from a renal cyst of an ADPKD patient[41] (Supplementary Fig. 17a). CRISPRi resulted in ~50% decrease of *MIR31HG* expression, confirming its enhancer activity for *MIR31HG* (Supplementary Fig. 17b). *CDKN2A* expression was not changed by CRISPRi of this region, suggesting that *CDKN2A* was not a direct target of that enhancer (Supplementary Fig. 17b). Although *MIR31HG* has been previously shown to regulate *CDKN2A* expression in a fibroblast cell line[35], the ~50% decrease in *MIR31HG* expression induced by CRISPRi may have been insufficient to alter *CDKN2A* levels in WT9-12 cells. Alternatively, the regulation of senescence-related genes in an immortalized cell line may not reflect regulation in a non-immortalized cell line. Despite this uncertainty, activation of this 5′ enhancer with *MIR31HG* upregulation in the ADPKD collecting duct is consistent with the notion that *MIR31HG* may promote cyst growth via suppression of cellular senescence.

We also analyzed the *GPRC5A cis*-coaccessibility network. This showed no difference in accessibility of the *GPRC5A* transcriptional start site across collecting duct subtypes. However, we identified a ~17 kb 5′ distal region that was differentially accessible in the PKD-CDC subpopulation that was predicted to interact in cis with the *GPRC5A* transcriptional start site (Fig. 8e and f, Supplementary Data 16). Analysis of the GPRC5A *cis*-coaccessibility network across kidney cell types (Supplementary Fig. 18) indicated that cis-coaccessibility between the 5′ distal region and the *GPRC5A* promoter was also present in PT, TAL, and IC clusters as well as PKD-CDC subset of CNT-PC. Our ability to detect cis-coaccessibility in PT despite the very small 5′ distal differentially accessible region in these cell types (Supplementary Fig. 19a) suggests a robust *cis*-coaccessibility between these two genomic regions. Detection of the *cis*-coaccessibility network in TAL and IC was consistent with mild upregulation of *GPRC5A* gene expression among distal nephron cell types in ADPKD kidneys (Supplementary Fig. 19b).

To determine whether this 5′ distal region has enhancer activity for *GPRC5A* gene expression, we also performed CRISPRi in WT9-12 cells (Fig. 8g)[15]. CRISPRi on the promoter region achieved a ~90% decrease of *GPRC5A* expression as expected, while targeting the 5′ distal region induced a 40–50% decrease, confirming its enhancer activity (Fig. 8g). CRISPRi on the promoter or the distal area unexpectedly slightly upregulated the expression of surrounding genes in WT9-12 cells. This slight upregulation of surrounding gene expression may be due to unspecified effects of CRISPRi on neighboring chromatin accessibility or possibly secondary effect of *GPRC5A* downregulation. Regardless of the cause, the results strongly support that the 5′ distal differentially accessible region has enhancer activity for the *GPRC5A* gene (Fig. 8e).

This 5′ distal enhancer has several binding motifs for cAMP responsive element binding protein 1 (CREB1) and retinoic acid receptors (RAR) as well as NF-κB (RELB) and TEAD family transcription factors (TEAD1-4), based on JASPAR 2018[42,43] (Enrichment score > 300, Supplementary Fig. 20). Aberrant activation of cAMP signaling has been linked to disease progression in ADPKD, and CREB1 mediates cAMP-dependent gene regulation. Recent lines of evidence suggest

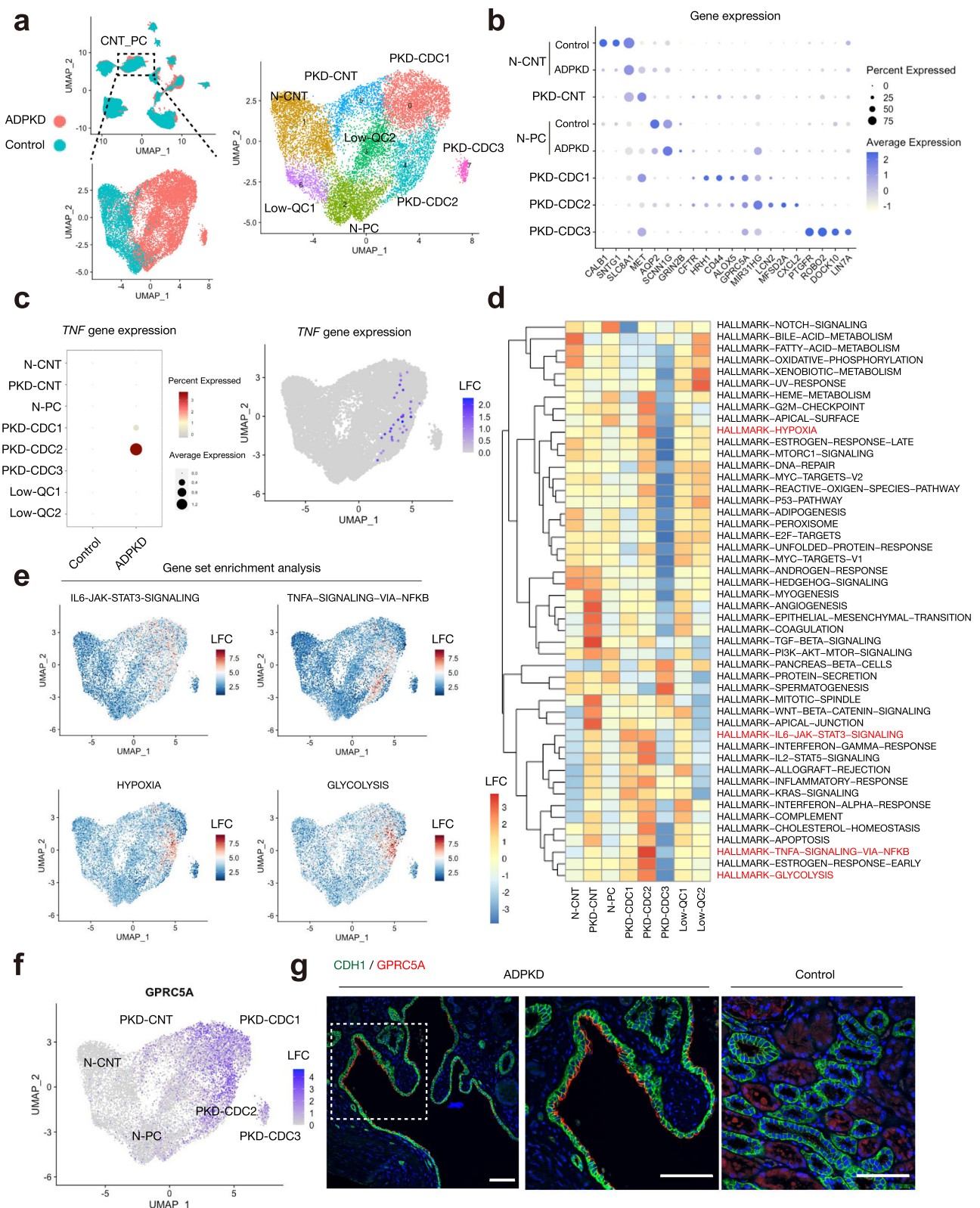

that the retinoic acid signaling pathway mediated by RAR in the collecting duct plays a protective role in kidney injury[44]. Interestingly, retinoic acid signaling was shown to be also activated by vasopressin in collecting duct cells[45], providing evidence of crosstalk between cAMP signaling and retinoic acid signaling pathways. The binding motifs for CREB1 and retinoic acid receptors were enriched in ADPKD cells (Fig. 8i, Supplementary Data 12), suggesting that these transcription factors were activated, potentially also inducing *GPRC5A* expression in

ADPKD. In agreement with this finding, the expression of retinol dehydrogenase 10 (*RDH10*), which is a rate-limiting enzyme of retinoic acid synthesis was maintained among PC subpopulations in ADPKD kidneys (Supplementary Fig. 21). Previous literature suggests that cAMP and retinoic acid stimulation upregulates GPRC5A expression in cancer cell lines[46,47]. To validate these findings, we treated WT9-12 cells with cAMP-inducing agent forskolin with or without all-trans retinoic acid, and we confirmed that cAMP and retinoic acid signaling increased

**Fig. 7 | Transcriptomic characterization of cyst-lining cells originated from collecting duct. a** Subclustering of CNT_PC on the UMAP plot of snRNA-seq dataset with annotation by disease condition (left) or subtype (right). N-CNT normal CNT, N-PC normal PC, PKD-CNT ADPKD-specific CNT, PKD-CDC ADPKD-specific collecting duct cells, LowQC low-quality cells. **b** Dot plot showing gene expression patterns of the genes enriched in each subpopulation. For N-PC and N-CNT, control and ADPKD data were individually shown. The diameter of the dot corresponds to the proportion of cells expressing the indicated gene and the intensity of the dot corresponds to average expression relative to all CNT_PC cells. **c** Dot plot showing *TNF* expression among subclusters (left). The diameter of the dot corresponds to the proportion of cells expressing the indicated gene and the density of the dot corresponds to average expression relative to all CNT_PC cells.

UMAP plot displaying *TNF* expression (right). The color scale represents a normalized log-fold-change (LFC). **d** Heatmap showing enrichment of hallmark gene-sets in each cell type in CNT_PC clusters. Source data are provided as a Source Data file. **e** UMAP displaying enrichment of genes upregulated by IL6 via STAT3 (upper left), genes regulated by NF-κB pathway in response to TNFα (upper right), genes upregulated in response to hypoxia (lower left) and genes encoding proteins involved in glycolysis and gluconeogenesis (lower right) in snRNA-seq dataset. **f** UMAP plot displaying *GPRC5A* gene expression in CNT_PC subtypes. The color scale represents a normalized LFC. **g** Representative immunofluorescence images of CDH1 (green) and GPRC5A (red) in the ADPKD (left and middle, *n* = 3) or control kidneys (right, *n* = 3). Scale bar indicates 50 μm.

*GPRC5A* expression in the kidney cell line (Fig. 8j). These results are consistent with the notion that cAMP drives expression of GPRC5A, potentially participating in proliferation of cyst cells.

## Discussion

We performed multimodal single-cell analysis on adult human ADPKD kidneys to untangle cellular complexity and dissect the molecular foundation of disease progression at single-cell resolution. Our analysis elucidates previously unrecognized cellular heterogeneity defined by single-cell gene expression and chromatin accessibility patterns in ADPKD kidneys.

Previous studies in mice have described a maladaptive repair cell state in proximal tubular cells. *Vcam1* + *Ccl2* + failed-repair cell states have been identified in snRNA-seq of mouse kidneys after ischemia reperfusion injury, and their transcriptomic signatures indicated NF-κB activation[21,22]. Furthermore, we recently identified a VCAM1 + subpopulation in PT of healthy human kidneys through multimodal single-cell analysis[12]. The transcriptomic signature of VCAM1 + PT was shown to be similar to mouse failed-repair PT population, suggesting that a small number of failed-repair PT cells also exist in human healthy kidneys. In this study, we found that most of the PT adopt a FR-PTC gene expression signature in ADPKD kidneys (Fig. 5). Furthermore, our subclustering analysis suggests previously unrecognized heterogeneity amongst FR-PTC cell states (Fig. 5b). Some PT subsets had lower *VCAM1* expression (PT-3/4) compared to the remaining (PT-1/2), although they expressed higher levels of other FR-PTC signature genes (Fig. 5c), suggesting that *VCAM1* is not a sole defining marker of the FR-PTC state, and that combinations of several markers may better classify damaged PT cell states. The heterogeneity of the failed-repair state is further reflected by the variability in correlation between ADPKD FR-PTC and mouse FR-PTC (Fig. 5d). For example, PT-3 strongly expressed some FR-PTC marker genes (*CREB5, TPM1, PROM1*[*CD133*], *TGFB2*) but did not express *VCAM1*, suggesting that this cell state may be transitioning either toward or away from VCAM1 + cell states. We also described heterogeneity of proinflammatory or profibrotic signaling among these subsets (Fig. 5g, h), implicating a potentially unique role of each cell state in CKD or cyst progression. The extent to which each PT subset may contribute to disease progression remains undefined. A better understanding of PT heterogeneity and how these states contribute to disease remains a major future challenge.

Although a large fraction of FR-PTC did not originate from cyst-lining cells but rather injured tubules, they still expressed proinflammatory (*CCL2*) or profibrotic (*TGFB2*) molecules (Fig. 5c), suggesting that they may be contributing to interstitial inflammation and fibrosis even though they are not cystic. CCL2 was previously shown to promote cystogenesis via recruitment of macrophages in model mice[48]. The role of TGFβ signaling in ADPKD progression is still controversial[49,50], although its profibrotic effect on the kidneys has been widely accepted[51]. Several lines of evidence indicate that ischemic injury promotes cystogenesis in a mouse model of ADPKD[52]. We speculate that FR-PTC accelerate cyst growth both after injury but also in the interstitial microenvironment made hypoxic by adjacent cyst

growth. Interestingly, FR-PTC also upregulated TGFβ receptor expression (Supplementary Fig. 11a), and ligand-receptor analysis suggested PT lineage as major target of TGFβ signaling in ADPKD (Fig. 4). An autocrine loop of TGFβ signaling was previously found to lead to aberrantly high levels of TGFβ2 through CREB1 and SMAD3 binding to the *TGFB2* promoter in glioblastoma[53]. This suggests that TGFβ signaling in FR-PTC may not only induce fibrosis but also maintain the FR-PTC cell state in CKD kidneys. These findings also suggested a potential therapeutic angle for cAMP-CREB1-TGFB2 axis in FR-PTC.

ADPKD is associated with various degrees of interstitial fibrosis, which is the final common pathway of CKD regardless of etiology[51]. Interstitial fibrosis has been shown to be an accelerator of disease progression in ADPKD, as well as other CKD[54]. Renal fibrosis is a complicated series of processes including tissue injury, inflammation, myofibroblast proliferation, and deposition of extracellular matrix in the tissue to cause irreversible tissue remodeling[51]. Many cell types and fibroblast subtypes are involved in fibrosis, although the heterogeneity of fibroblasts in ADPKD kidneys and the roles of each fibroblast subtypes have not been elucidated. Here, we identified two fibroblast subtypes predominant in ADPKD kidneys: *ACTA2* + myofibroblasts and *FGF14* + *IL6* + fibroblasts (Fig. 6). Interestingly, we found the latter fibroblast subtype expressed *IL6* at the highest level among all ADPKD kidney cell types (Fig. 4, Supplementary Fig. 12a), suggesting a role in the inflammatory microenvironment. Our study is limited by the fact that the ADPKD kidney samples were from end stage disease, and we do not have sufficient evidence if *FGF14* + *IL6* + fibroblasts were specific to ADPKD.

Principal cells of the collecting duct have been proposed as the main origin of cyst in ADPKD[1,55,56]. We identified a GPRC5A + CDH1 + PC lineage (PKD-CDC1) in cyst-lining cells (Fig. 7f, g). This subpopulation also differentially expresses *MET*, which mediated HGF-dependent mTORC activation in a mouse model of ADPKD[30]. Interestingly, *Gprc5a* was previously identified as one of the *Pkd1*-mutant signature genes that was upregulated in conditional *Pkd1*-KO mouse kidneys, suggesting a role for this gene in cystogenesis in mice[57]. GPRC5A expression is induced by hypoxia in a cancer cell line, subsequently activating YAP/TAZ to promote survival of the cells in hypoxic conditions[33]. YAP/TAZ-dependent gene regulation is mediated by TEAD family transcription factors[58]. In agreement with this, TEAD3 activity was increased in the PC lineage in ADPKD kidneys in snATAC-seq (Fig. 8c) Furthermore, another YAP/TAZ activator ROR1 was also expressed in GPRC5A + cysts (Fig. 8d). ROR1 is a receptor tyrosine kinase that is overexpressed in many types of cancers to activate non-canonical WNT signaling pathway. It is tempting to speculate that upregulation of GPRC5A and ROR1 in cyst-lining cells might drive proliferation in the hypoxic milieu in ADPKD kidneys. Given that GPRC5A protein was previously identified in urinary exosomes[59,60], measurement of GPRC5A in urinary exosomes from ADPKD patients could serve as a biomarker for disease progression. Another PC subpopulation (PKD-CDC2) expresses *LCN2* that was shown to be expressed in cyst cells of a mouse model and human samples[31]. PKD-CDC2 was also shown to be

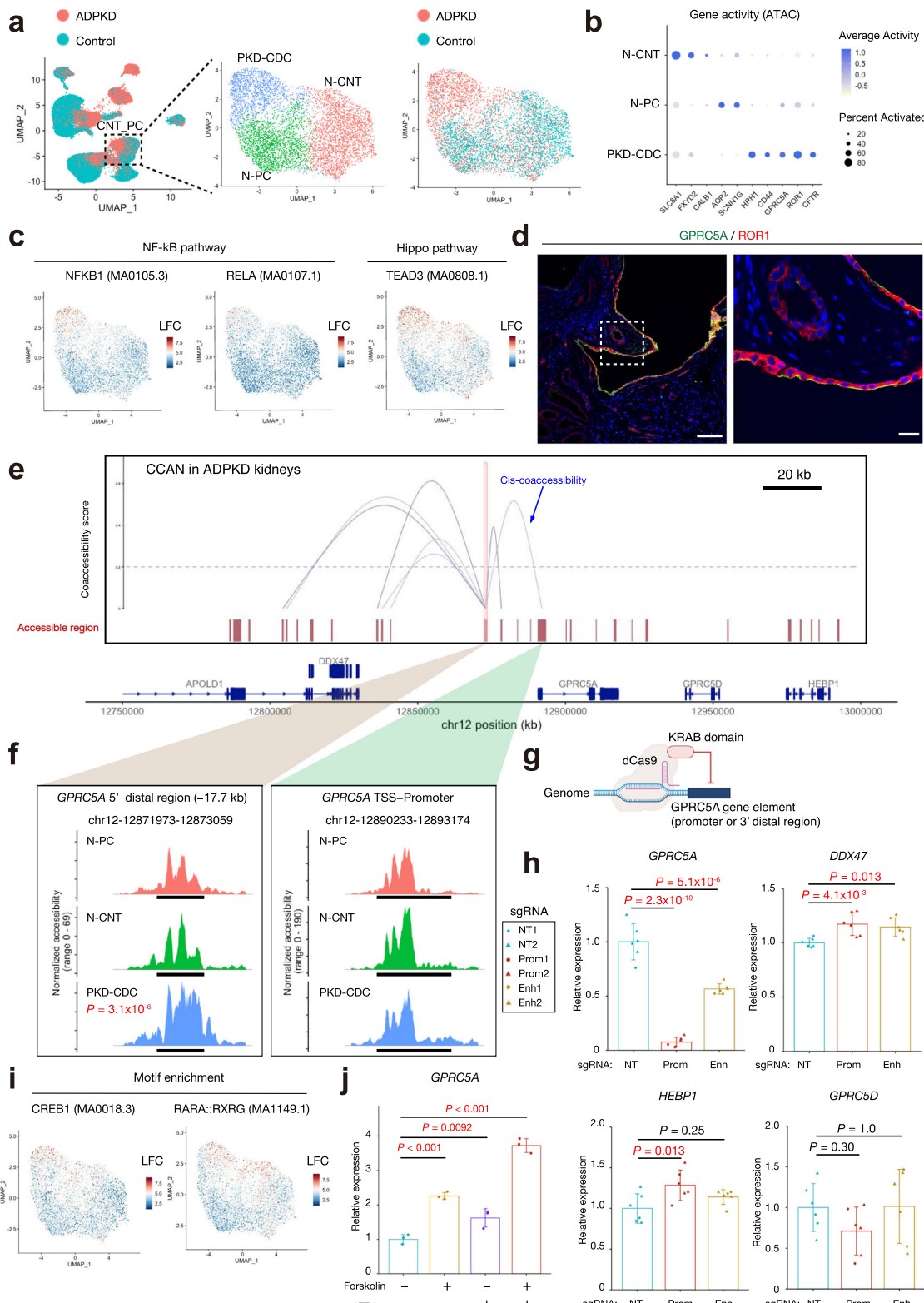

the major source of TNF signaling in ADPKD kidneys, suggesting a significant proinflammatory role (Figs. 4 and 7c).

We leveraged snATAC-seq data to predict an enhancer element for GPRC5A expression in ADPKD kidneys (Fig. 8), and validated that the enhancer regulates GPRC5A expression by CRISPR interference (Fig. 8h). Given that promoter accessibility was not changed between ADPKD and control cells (Fig. 8f), this differentially accessible enhancer may be responsible for upregulation of *GPRC5A* gene expression in ADPKD PC. We observed RAR and CREB1 binding sites in that enhancer

region (Supplementary Fig. 20), and these transcription factor binding motifs were enriched in accessible regions of PC lineage in ADPKD kidneys (Fig. 8i), suggesting that *GPRC5A* is regulated by retinoic acid signaling and cAMP signaling pathways. Indeed, we confirmed that *GPRC5A* expression was regulated by these signaling pathways in human ADPKD cyst cell line (Fig. 8j). Several lines of evidence suggest retinoic acid signaling pathway in collecting duct plays a protective role against kidney injury[44]. Aberrant activation of the retinoic acid signaling pathway may promote cyst growth and interstitial

**Fig. 8 | Multimodal approach revealed epigenetic alterations in ADPKD cyst cells. a** Subclustering of CNT_PC on the UMAP plot of the snATAC-seq dataset with annotation by subtype (left) or disease condition (right). **b** Dot plot showing gene activity patterns of the genes enriched in each of CNT_PC subpopulations. The diameter of the dot corresponds to the proportion of cells with detected activity of indicated gene and the intensity of the dot corresponds to average gene activity relative to all CNT_PC nuclei. **c** UMAP plot showing enrichment of transcription factor binding motifs for NF-κB pathway; NFKB1 (left) and RELA (middle), or Hippo pathway; TEAD3 (right). The color scale represents a normalized log-fold-change (LFC). **d** Representative immunofluorescence images of ROR1 (red) and GPRC5A (green) in the ADPKD kidneys ($n = 3$). Scale bar indicates 50 μm (left) or 10 μm (right). **e** Cis-coaccessibility network (CCAN, gray arcs) among accessible regions (red boxes) around the *GPRC5A* locus is shown. **f** Fragment coverage (frequency of Tn5 insertion) around TSS (middle right, chr12:12890233–12893174) or 5′ distal differentially accessible region (middle left, chr12:12871973–12873059) are shown (peak ±1 Kb). Bonferroni-adjusted *p*-values were used to determine significance for differential accessibility. **g** Graphic methodology showing CRISPR interference. Schematic was created with BioRender. **h** RT and real-time PCR analysis of mRNAs for *GPRC5A* or its surrounding genes (*DDX47*, *HEBP1*, and *GPRC5D*) in WT9-12 cells with CRISPR interference targeting the promoter (Prom) or 5′ distal potential enhancer (Enh) for *GPRC5A* gene. NT nontargeting control. Each group consists of $n = 6$ data (2 sgRNAs with 3 biological replicates). **i** UMAP plot showing enrichment of transcription factor binding motifs for CREB1 (left) or retinoic acid receptor (RARA::RXRG, right). The color scale represents a normalized LFC. **j** RT and real-time PCR analysis of mRNAs for *GPRC5A* in WT9-12 cells treated with forskolin (10 μM) with or without all-trans retinoic acid (ATRA, 1 μM) for 6 h ($n = 3$ biological replicates). Bar graphs represent the mean and error bars are the s.d. One-way ANOVA with post hoc Dunnett's multiple comparisons test. Source data are provided as a Source Data file (**h**, **j**).

remodeling in ADPKD kidneys. This pathway was also found to be activated by vasopressin[45], a pathway whose inhibition slows progression in ADPKD.

The long non-coding RNA *MIR31HG* was overexpressed both in PKD-CDC1 and PKD-CDC2, although *MIR31HG* transcripts were more abundant in PKD-CDC2 (Fig. 7b). *MIR31HG* regulates HIF1A-dependent transcription[34]. We hypothesize that *MIR31HG* may be driving expression of highly expressed glycolysis genes in PKD-CDC2. Furthermore, *MIR31HG* is upregulated in oncogene-induced cellular stress through 5′ distal enhancer activation, and in that context suppressed expression of *CDKN2A* via recruitment of polycomb group proteins[35]. This same 5′ distal enhancer was differentially accessible in ADPKD (Supplementary Fig. 15c). *CDKN2A* was highly expressed in PC subpopulations in ADPKD, especially in PKD-CDC1 (Supplementary Fig. 15b), implicating cell stress-induced senescence in these cells. *MIR31HG* expression in PKD-CDC1 or PKD-CDC2 may have a role in cyst growth through suppression of cellular senescence, given the lines of evidence suggest senescence delays cyst growth[36]. Collectively, these findings shed light on heterogeneity of cyst-lining cells in ADPKD.

In summary, we performed multimodal single-cell analysis on ADPKD kidneys for more precise delineation of disease-specific cell states. Our study is limited by the fact that the ADPKD kidney samples were from end stage disease, and that the controls were from individuals with normal kidney function. However, ESKD samples are the only human source of ADPKD tissue available. In the future, application of spatially resolved transcriptomics on human ADPKD kidneys should help to further dissect intercellular communication in the ADPKD microenvironment. Our single-cell multimodal analysis of human ADPKD kidney redefines cellular heterogeneity in ADPKD kidneys, and provides a single-cell multiomic foundation on which to base future efforts to develop better diagnostic and therapeutic approaches for ADPKD.

## Methods
### Tissue procurement
ADPKD kidney cortical cup samples were obtained from patients undergoing simultaneous native nephrectomy and living donor kidney transplantation at the University of Maryland Medical Center (Baltimore, MD). The Maryland PKD Research and Translation Core Center, in a unique relationship with a team of transplant surgeons at the University of Maryland Medical Center, has developed a protocol for the PKD center to receive intact nephrectomized kidneys as soon as they are removed from the transplant patient, at the operation room. The PKD researcher then immediately begins the dissection and collection of samples. The kidneys are cooled on ice as soon as they are removed from the patient and during the entire dissection and sample preparation process. The samples investigated here were collected from the base (cup) of large superficial cysts (Supplementary Fig. 1) with portions fixed and portions flash frozen. By design, each sample contained the epithelial wall of one large cyst, but previous analysis demonstrated that the samples also contained significant other cysts of all sizes. We are not able to quantitate percentage of various cyst sizes since each sample is unique.

All participants provided written informed consent for participation and tissue donation. The human subjects protocol was approved by the Institutional Review Board of the University of Maryland, Baltimore. Non-tumor kidney cortex samples for controls were obtained from patients undergoing partial or radical nephrectomy for renal mass at Brigham and Women's Hospital (Boston, MA) under an established Institutional Review Board protocol approved by the Mass General Brigham Human Research Committee. The multimodal single-cell dataset generated from control kidneys (10X Genomics Chromium Single Cell 5′ v2 chemistry and 10X Genomics Chromium Single Cell ATAC v1) were already published[12]. Control kidneys were newly processed to obtain snRNA-seq libraries with 10X Genomics Chromium Single Cell 3′ v3 chemistry for this manuscript. All participants provided written informed consent in accordance with the Declaration of Helsinki. Samples were frozen or retained in paraffin blocks for future studies.

### Nuclear dissociation for library preparation
For snATAC-seq, nuclei were isolated with Nuclei EZ Lysis buffer (NUC-101; Sigma-Aldrich) supplemented with protease inhibitor (5892791001; Roche). Samples were cut into <2 mm pieces, homogenized using a Dounce homogenizer (885302−0002; Kimble Chase) in 2 ml of ice-cold Nuclei EZ Lysis buffer, and incubated on ice for 5 min with an additional 2 ml of lysis buffer. The homogenate was filtered through a 40 μm cell strainer (43−50040−51; pluriSelect) and centrifuged at $500 \times g$ for 5 min at 4 °C. The pellet was resuspended, washed with 4 ml of buffer, and incubated on ice for 5 min. Following centrifugation, the pellet was resuspended in Nuclei Buffer (10× Genomics, PN-2000153), filtered through a 5 μm cell strainer (43-50005-03, pluriSelect), and counted. For snRNA-seq preparation, the RNase inhibitors (Promega, N2615 and Life Technologies, AM2696) were added to the lysis buffer, and the pellet was ultimately resuspended in nuclei suspension buffer (1× PBS, 1% bovine serum albumin, 0.1% RNase inhibitor). Subsequently, 10X Chromium libraries were prepared according to manufacturer protocol.

### Single-nucleus RNA sequencing and bioinformatics workflow
Eight ADPKD and five control snRNA-seq libraries were obtained using 10X Genomics Chromium Single Cell 3′ v3 chemistry following nuclear dissociation. A target of 10,000 nuclei were loaded onto each lane. The cDNA for snRNA libraries was amplified for 15 cycles (Sample index PCR). Libraries were sequenced on an Illumina Novaseq instrument and counted with cellranger v6.0.0 with --include-introns

argument using GRCh38. The read configuration for the libraries was 2 × 150 bp paired-end. A mean of 408,304,417 reads (s.d. = 342,469,382, control) or 358,474,996 reads (s.d. = 59,365,096, ADPKD) were sequenced for each snRNA library corresponding to a mean of 33,629 reads per cell (s.d. = 23,330, control) or 44,799 reads per cell (s.d. = 32,727, ADPKD, Supplementary Table 4). The mean sequencing saturation was 47.6 ± 13.4% (control) or 53.0 ± 15.4% (ADPKD, Supplementary Table 5). The mean fraction of reads with a valid barcode (fraction of reads in cells) was 53.9 ± 6.8% (control) or 36.9 ± 5.9% (ADPKD, Supplementary Table 5).

The output of cellranger (filtered_gene_bc_matrix) were processed through Seurat v4.0.0[16]. Ambient RNA contamination was corrected for each dataset by SoupX v1.5.0[61] with automatically calculated contamination fraction. Each of datasets was then processed to remove low-quality nuclei (nuclei with top 5% and bottom 1% in the distribution of feature count or RNA count, or those with %Mitochondrial genes >0.25). Heterotypic doublets were identified with DoubletFinder v2.0.3[27] assuming 8% of barcodes represent heterotypic doublets), and resultant estimated doublets were to be removed after merging datasets. The datasets from ADPKD or control kidneys were integrated in Seurat using the IntegrateData function with anchors identified by FindIntegrationAnchors function (Supplementary Fig. 2a, 3a). Subsequently, the doublets and low-QC clusters were removed for these datasets (Supplementary Fig. 2b, 3b). The ADPKD and control datasets were integrated with batch effect correction with Harmony v1.0[17] using RunHarmony function on assay RNA in Seurat (Fig. 1). Then, there was a mean of 8127 ± 1692 nuclei in control or 7759 ± 3377 nuclei in ADPKD per snRNA-seq library. The number of unique molecular identifiers (UMI) per nucleus was a mean of 3536 ± 1914 in control or 2346 ± 1274 in ADPKD. The number of detected genes per nucleus was a mean of 2222 ± 803 genes in control or 1743 ± 573 genes in ADPKD. %Mitochondrial genes in a nucleus was 0.027 ± 0.050% in control or 0.0077 ± 0.029% in ADPKD (Supplementary Fig. 4). Clustering was performed by constructing a KNN graph and applying the Louvain algorithm. Dimensional reduction was performed with UMAP and individual clusters were annotated based on expression of lineage-specific markers (Fig. 1). The final snRNA-seq library contained 62,073 nuclei from ADPKD and 40,637 nuclei from control kidneys, and represented all major cell types within the kidney cortex (Supplementary Table 2 and Fig. 1). Differential expressed genes among cell types were assessed with the Seurat FindMarkers function for transcripts detected in at least 20% of cells using a log-fold-change threshold of 0.25. Differential expressed genes between ADPKD and control cells in each cell type were assessed for transcripts detected in at least 10% of cells using a log-fold-change threshold of 0.25 (Supplementary Data 1–3). Bonferroni-adjusted *p*-values were used to determine significance at an FDR < 0.05. Gene expressions were visualized with FeaturePlot (UMAP), VlnPlot (violin plot) or DotPlot (dot plot) function on Seurat or complex_dotplot_single function (dot plot) on R package plot1cell (v0.0.0.9000, https://github.com/TheHumphreysLab/plot1cell).

### Single-nucleus ATAC sequencing and bioinformatics workflow
Eight ADPKD kidney snATAC-seq libraries were obtained using 10X Genomics Chromium Single Cell ATAC v1 chemistry following nuclear dissociation. Five control snATAC-seq libraries (Control 1–5) were prepared and published in a prior study (GSE151302)[12]. Libraries were sequenced on an Illumina Novaseq instrument and counted with cellranger-atac v1.2 (10X Genomics) using GRCh38. The read configuration was 2 × 150 bp paired-end. Sample index PCR was performed at 13 cycles. A mean of 334,652,440 reads were sequenced for each snATAC library (s.d. = 95,862,297) corresponding to a median of 21,671 fragments per cell (s.d. = 11,946, Supplementary Table 4). The mean sequencing saturation for snATAC libraries was 31.6 ± 9.7% and the mean fraction of reads with a valid barcode was

95.2 ± 3.9% (Supplementary Table 6). The libraries from control and ADPKD kidneys were aggregated with cellranger-atac v1.2.0. Subsequently, the aggregated dataset (filtered_peak_bc_matrix) was processed with Seurat v4.0.0 and its companion package Signac v1.1.1[11]. Low-quality cells were removed from the aggregated snATAC-seq library (subset the high-quality nuclei with peak region fragments >1000, peak region fragments <12000, %reads in peaks >15, blacklist ratio <0.005, nucleosome signal <3, and TSS enrichment >2). Latent semantic indexing was performed with term-frequency inverse-document-frequency (TFIDF) followed by singular value decomposition (SVD). A KNN graph was constructed to cluster cells with the Louvain algorithm. Batch effect was corrected with Harmony[17] using the RunHarmony function in Seurat. A gene activity matrix was constructed by counting ATAC peaks within the gene body and 2 kb upstream of the transcriptional start site using protein-coding genes annotated in the Ensembl database. The gene activity matrix was log-normalized.

For label transfer, the above snATAC-seq Seurat object was divided to control and ADPKD kidney dataset, and label transfer was performed on each of the control and ADPKD kidney dataset, using filtered control and ADPKD snRNA-seq dataset (Supplementary Figs. 2b, 3b), respectively. FindTransferAnchors and TransferData functions were used for label transfer, according to instructions (https://satijalab.org/signac/)[11]. After label transfer, the control and ADPKD snRNA-seq datasets were filtered using an 80% confidence threshold for low-resolution cell-type assignment to remove heterotypic doublets (Supplementary Fig. 7). The filtered control and ADPKD snATAC-seq objects were merged and reprocessed with TFIDF and SVD. Subsequently, the dataset was processed for batch effect correction with Harmony[17], clustering and cell-type annotation based on lineage-specific gene activity (Fig. 2b–d). The final snATAC-seq library contained a total of 128,008 peak regions among 50,986 nuclei (33,621 nuclei for control and 17,365 nuclei for ADPKD) and represented all major cell types within the kidney cortex (Supplementary Table 3). The number of fragments in peaks per nucleus was a mean of 5560 ± 2325 in control or 4186 ± 2196 in ADPKD, %Fragments per nucleus in reads was a mean of 57.0 ± 10.8% in control or 38.6 ± 11.8% in ADPKD. Fraction of reads in peaks, number of reads in peaks per cell and ratio of reads in genomic blacklist regions per cell for each patient were shown in Supplementary Fig. 4. Differential chromatin accessibility among cell types was assessed with the Seurat FindMarkers function for peaks detected in at least 20% of cells with a likelihood ratio test and a log-fold-change threshold of 0.25. Differential chromatin accessibility between ADPKD and control cells in each cell type was assessed for peaks detected in at least 10% of cells using a log-fold-change threshold of 0.25 (Supplementary Data 4–6). Differential gene activities among cell types or between ADPKD and control cells in each cell type were assessed with the Seurat FindMarkers function with a log-fold-change threshold of 0.25 (Supplementary Data 7-9). Bonferroni-adjusted *p*-values were used to determine significance at an FDR < 0.05.

### Subclustering of each cell type
For subclustering of snRNA-seq data, the target cell type was extracted based on the annotations on the integrated dataset (Fig. 1b). Subsequently, the target cell type was further filtered based on the annotations on each of control and ADPKD datasets (Supplementary Fig. 2b, Supplementary Fig. 3b) to extract the target cell type with high confidence. The batch effects in the extracted target cell type was corrected with Harmony v1.0[17]. Subsequently, clustering was performed by constructing a KNN graph and applying the Louvain algorithm. Dimensional reduction was performed with UMAP. For snATAC-seq data, the target cell type was extracted based on the annotations on the integrated dataset (Fig. 2b). Subsequently, clustering was performed by constructing a KNN graph and applying the Louvain

algorithm. Subclustering in snATAC-seq dataset was performed with the use of Harmony embedded on the whole dataset. Dimensional reduction was performed with UMAP. FindMarkers function was used to assess differentially expressed genes, differentially accessible regions or differentially enriched transcription factor binding motifs with a log-fold-change threshold of 0.25 (Supplementary Data 13–17). Bonferroni-adjusted $p$-values were used to determine significance at an FDR < 0.05.

## Estimation of transcription factor activity from snATAC-seq data

Transcription factor activity was estimated using the integrated snATAC-seq dataset and chromVAR v1.10.0[10]. The positional weight matrix was obtained from the JASPAR2018 database[42]. Cell-type-specific chromVAR activities were calculated using the Run-ChromVAR wrapper in Signac v1.1.1 and differential activity was computed with the FindMarkers function with mean.fxn = rowMeans and fc.name = avg_diff. (Log-fold-change > 0.25 for comparison among cell types and Log-fold-change > 0.1 for comparison between ADPKD and control in each cell type, Supplementary Data 10–12). The chromvar activity in each transcription factor on the whole dataset was shown with FeaturePlot function with max.cutoff = q99 and min.cutoff = q1 or VlnPlot function (Fig. 3c–e)

## Generation of cis-coaccessibility networks with Cicero

Cis-coaccessibility networks (CCAN) were predicted using Cicero v1.3.4.11 according to instructions provided on GitHub (https://cole-trapnell-lab.github.io/cicero-release/docs_m3/)[9]. Briefly, the ADPKD data was extracted from integrated snATAC-seq dataset and converted to cell dataset (CDS) objects using the make_atac_cds function. The CDS object was processed using the detect_genes() and estimate_size_factors() functions with default parameters prior to dimensional reduction and conversion to a Cicero CDS object. ADPKD-specific Cicero connections were obtained using the run_cicero function with default parameters. CCAN was visualized with plot_connections function with coaccess_cutoff = .2 (for whole ADPKD dataset, Fig. 8e, Supplementary Fig. 15c) or .05 (for each ADPKD cell type, Supplementary Fig. 18).

## Single-cell gene enrichment analysis on snRNA-seq data

Single-cell geneset enrichment analysis was performed with the VISION v2.1.0 R package according to instructions provided on GitHub (https://github.com/YosefLab/VISION)[18], using Hallmark genesets obtained from the Molecular Signatures Database v7.4 distributed at the GSEA Web site. The resultant enrichment scores were incorporated into Seurat object, and visualized on UMAP using FeaturePlot function with max.cutoff = q99 and min.cutoff = q1 (Fig. 3b). The heatmaps were generated with pheatmap v1.0.12 from geneset enrichment scores averaged in each cell type (Fig. 3a) or subtype (Figs. 5g and 7d) with AverageExpression function.

## Ligand-receptor analysis on snRNA-seq data

Ligand-receptor analysis was performed with CellChat v1.1.3 R package according to instructions provided on GitHub (https://github.com/sqjin/CellChat)[20]. Briefly, the ADPKD data was extracted from integrated snRNA-seq dataset, and the CellChat object was generated using the createCellChat function. The object was then preprocessed (identifyOverExpressedGenes, identifyOverExpressedInteractions and projectData functions), and communication probability was computed with computeCommunProb function with type = truncatedMean and trim = 0.001. The cell–cell communication was inferred at a signaling pathway level with computeCommunProbPathway function. The data were visualized with netVisual_heatmap, netVisual_aggregate or netAnalysis_contribution function.

## Deconvolution of published ADPKD data

The human microarray dataset GSE7869 was retrieved from the Gene Expression Omnibus database (GEO)[28]. The dataset was deconvoluted with ADPKD snRNA-seq dataset using CIBERSORTx executables v1.0[29] according to instructions provided on CIBERSORTx website (https://cibersortx.stanford.edu). Briefly, cell-type fraction (CIBERSORTx Fractions) was predicted with --single_cell TRUE --rmbatchSmode TRUE --perm 100. Cell-type-specific expression purification at high resolution (CIBERSORTx HiRes) was performed with the signature matrix generated in the cell-type fraction prediction.

## Pathway analysis on snRNA-seq data with PROGENy

PROGENy (v. 1.15.3, https://saezlab.github.io/progeny/) was applied to the snRNA-seq Seurat object with progeny function[25]. Each pathway was scaled to have a mean activity of 0 and a standard deviation of 1. The PROGENy pathway activity scores were computed on the scRNA-seq data, and then the different cell populations were characterized based on these scores. The different pathway activities for the different cell populations were then plotted as heatmaps.

## Correlation between ROR1 expression and cyst size

The human microarray dataset GSE7869 was retrieved from the Gene Expression Omnibus database (GEO)[28]. The dataset contained $n = 3$ non-ADPKD control kidneys, $n = 5$ of minimally cystic tissue, $n = 5$ of small-sized renal cysts, $n = 5$ of medium-sized renal cysts, and $n = 3$ of large-sized renal cysts. The renal cyst samples were obtained from five $PKD1$-mutant polycystic kidneys. To visualize the correlation of ROR1 in renal cysts, the expression level of ROR1 in units of normalized signal intensity was plotted against the grouped cyst size.

## Correlation of gene expression between human ADPKD and mouse kidney dataset

The PT lineage was extracted from published snRNA-seq dataset for mouse kidneys with ischemia reperfusion injury (GSE139107)[21], and the highly variable genes were identified with FindVariableFeatures function (nfeatures = 2000) in Seurat. Subsequently, the dataset was converted to human annotations using biomaRt and ensembl. The highly variable genes that also exist in human dataset after orthologous mouse-human lift over (biomaRt) were selected (1648 genes). These highly variable genes were analyzed with Pearson correlation (cor function). The resultant Pearson correlation coefficients between PT subtypes in IRI mice and those of human dataset were shown on a heatmap (pheatmap, Fig. 5d).

## Identification of transcription factor binding motifs in *GPRC5A* enhancer

Identification of transcription factor binding motifs in a *GPRC5A* enhancer was performed on UCSC genome browser[43] with TFBS predictions in *Homo sapiens* (hg38) in the JASPAR CORE vertebrates collection[42]. Minimum score was set to 300 which corresponds to *P*-value of 0.001.

## Immunofluorescence studies

Deparaffination of tissue samples was performed by immersing glass slides into coplin jars with xylene and ethanol (5 min in 100% xylene, 5 min in 100% xylene, 5 min in 100% ethanol, 5 min in 95% ethanol, 5 min in 70% ethanol, 5 min in distilled water and 5 min in distilled water). Following the last wash, slides were placed in antigen retrieval solution (Vector H-330). Samples were incubated in a pressure cooker (Prestige Medical Classic 2100 series). Following this incubation, samples were allowed to cool to room temperature, and washed with distilled water. Samples were treated with 2–3 drops of Image-iT FX Signal Enhancer (Molecular Probes; 136933) for 15 min with rotation at room temperature, and then blocked in Blocking Media [1% BSA (Roche; 03 116 956001), 0.1% Triton X-100 (Sigma; T8787), 0.1%

sodium azide (Sigma; S28032) in PBS] for another 15 min with rotation at room temperature. Primary antibody was added in Blocking Media [rabbit Anti-GPRC5A (Sigma; SAB4503536; 1:100), goat Anti-ROR1 (Abcam; Ab111174; 1:125), mouse Anti-E-Cadherin (BD Transduction; 610182; 1:200, lotus tetragonolobus lectin (LTL) (Vector Labs; B-1325; 1:100), Anti-VCAM1 (Abcam; ab134047; 1:200), Anti-Cubilin (R&D Systems AF3700; 1:200), Anti-LRP2 (Abcam, ab76969; 1:200)] and incubated overnight in a humidifier chamber at 4 °C. The next day, slides were quickly washed three times in PBS. Secondary antibodies [Donkey Anti-Goat (Invitrogen; A11057; 1:200), Donkey Anti-Mouse (Invitrogen; A21202; 1:200), Donkey Anti-Rabbit (Invitrogen; A10042 or Jackson ImmunoResearch; 711-545-152; 1:200), conjugated Streptavidin (Invitrogen; S21374; 1:200)] were added in Blocking Media and incubated at room temperature, in the dark for 1 h. Slides were then again quickly washed in PBS, incubated with DAPI (Invitrogen; D1306; 1:1000) in PBS for 5 min, and washed finally with PBS two more times for 5 min each. Following washes, coverslips were mounted onto glass slides with Prolong Gold Antifade Mountant (Invitrogen; P3690), and sealed 16 h later with nail polish. For co-staining with Anti-VCAM1 (rabbit monoclonal; Abcam; ab134047; 1:200) and Anti- LRP2 (rabbit polyclonal; Abcam; ab76969; 1:200), tissue was first stained with Anti-VCAM1, according to the aforementioned protocol. Following incubation with the secondary antibody (Invitrogen; A21206: 1:200), tissue was washed three times with 1× PBS for 5 min each and two additional times (5 min each) with 1× PBS with 0.05% Tween20 pH 7.4 (Sigma P3563). Then, the Anti-LRP2 was incubated overnight as per standard protocol. Tissue was then developed with secondary antibody (Invitrogen; A10042; 1:200), washed and mounted as described above. Imaging was performed on a Nikon Eclipse Ti Confocal at ×10 and ×20 objective and processed using Nikon Elements-AR and FIJI (Version 2.0.0-rc-68/1.52k). Imaging conditions (exposure time, laser intensity, etc) and processing (background subtraction, color balance, etc) were optimized for each antibody and maintained across samples.

### Cell culture

HEK293T cells (ATCC; CRL-3216) and WT9-12 cells (ATCC, CRL2833) were maintained in a humidified 5% $CO_2$ atmosphere at 37 °C in Dulbecco's modified Eagle's medium (DMEM, Gibco; 11965092) supplemented with 10% fetal bovine serum (Gibco; 10437028) and antibiotics. The dishes for WT9-12 cells were coated with bovine collagen I (R&D Systems, 3442-005-01). At 50–60% confluency, WT9-12 cells were treated with or without 10 μM of forskolin (Selleck Chemicals; S2449) and/or 1 μM of retinoic acid (Millipore Sigma; R2625) for 6 h.

### CRISPR interference

We designed small guide RNA (sgRNA) targeting *GPRC5A* promoter or the 5′ distal region co-accessible to that promoter with CHOPCHOP (https://chopchop.cbu.uib.no/). These sgRNAs and two nontargeting control sgRNAs were inserted into downstream of the U6 promoter on the dCas9-KRAB repression plasmid (pLV hU6-sgRNA hUbC-dCas9-KRAB-T2a-Puro, Addgene; 71236, a gift from Charles Gersbach)[15] with golden gate assembly. The sgRNA sequence which we used in the present study is listed in the Supplementary Table 7. Single strand oligonucleotides were purchased from Integrated Technology (IDT) and, sense and anti-sense oligonucleotides were annealed. Cloning with Golden gate assembly was performed with Esp3I restriction enzyme (NEB, R0734L) and T4 DNA ligase (NEB, M0202L) on a thermal cycler repeating 37 °C for 5 min and 16 °C for 5 min for 60 cycles, followed by transformation to NEB 5-alpha Competent E. coli (NEB, C2987H) as manufacturer's instruction. The cloned lentiviral vectors were purified with mini high-speed plasmid kit (IBI Scientific; IB47102), and sgRNA insertion was confirmed with Sanger sequencing by GENEWIZ.

To generate lentivirus, HEK293T cells were seeded at $6.0 \times 10^5$ cells per well on 6-well tissue culture plates 16 h before transfection.

Then, cells were transfected with 1.5 μg of psPAX2 (Addgene; 12260, a gift from Didier Trono), 0.15 μg of pMD2.G (Addgene; 12259, a gift from Didier Trono) and 1.5 μg of dCas9-KRAB repression plasmid per well by Lipofectamine 3000 transfection reagent (Invitrogen; L3000015) as the manufacturer's instructions. Culture media were changed to DMEM supplemented with 30% FBS 24 h after transfection. Lentivirus-containing supernatants were collected 24 h later, and they were filtered with 0.45 μm PVDF filters (CELLTREAT; 229745). The resultant supernatants were immediately used for lentiviral transduction. WT9-12 cells were seeded at $5.0 \times 10^4$ cells per well on 6-well tissue culture plates 16 h before transfection. The media on WT9-12 cells was then changed to the fresh lentiviral supernatants supplemented with polybrene (5 μg/ml, Santa Cruz Biotechnology; sc-134220) and cultured for 24 h. Subsequently, WT9-12 cells were cultured in DMEM with 10% FBS and puromycin (1 μg/ml, invivogen; ant-pr-1) for 72 h.

### Quantitative PCR

RNA from WT9-12 cells was extracted using the Direct-zol MicroPrep Plus Kit (Zymo) following the manufacturer's instructions. The extracted RNA (1–2 μg) was reverse transcribed using the High-Capacity cDNA Reverse Transcription Kit (Life Technologies). Quantitative PCR was carried out in the BioRad CFX96 Real-Time System using iTaq Universal SYBR Green Supermix (Bio-Rad). Expression levels were normalized to *GAPDH*, and the data were analyzed using the 2-ΔΔCt method. The following primers were used: *GAPDH*: Fw 5′- GACAGTCAGCCGC ATCTTCT −3′; Rv 5′- GCGCCCAATACGACCAAATC −3′; *GPRC5A*: Fw 5′- ATGGCTACAACAGTCCCTGAT −3′; Rv 5′- CCACCGTTTCTAGGACGA TGC −3′; *DDX47*: Fw 5′- GCACCCGAGGAACACGATT −3′; Rv 5′- TCCATCCCAACTGGTCACAAG −3′; *HEBP1*: Fw 5′- TTGGCAGGTCCT AAGCAAAGG −3′; Rv 5′- CTTCCCGTAGAGCCTCATCC −3′; *GPRC5D*: Fw 5′- CTGCATCGAGTCCACTGGAG −3′; Rv 5′- AAGAGTAGCAGAATTGT GACCAC −3′; *MIR31HG*: Fw 5′- CGCTTCTGTCCTCCTACTCG-3′; Rv 5′- ACAAGCAGACCCTTGGAATG −3′; *CDKN2A*: Fw 5′- CCCAACGCACCG AATAGTTA-3′; Rv 5′- ACCAGCGTGTCCAGGAAG −3′.

### Statistical analysis

No statistical methods were used to predetermine sample size. Experiments were not randomized and investigators were not blinded to allocation during library preparation, experiments or analysis. Predicted FIB frequency (Fig. 6e) or quantitative PCR data (Fig. 8, Supplementary Fig. 17) are presented as mean ± s.d. and were compared between groups with two-sided Student's *t*-test (Supplementary Fig. 17) or one-way ANOVA with post hoc Dunnett's multiple comparisons test (Figs. 6e and 8h, j). A *P*-value of <0.05 was considered statistically significant.

### Reporting summary

Further information on research design is available in the Nature Research Reporting Summary linked to this article.

## Data availability

All the data for this manuscript are publicly available. The sequencing data generated in this study have been deposited in GEO database under accession code GSE185948. Previously published snATAC-seq data for five control kidneys are available in GEO (GSE151302). Public data repositories used for our analyses include Ensembl http://useast.ensembl.org., Genome UCSC browser http://genome.ucsc.edu., and JASPAR http://jaspar.genereg.net. Gene expression, ATAC peaks, and gene activities for each cell type are also available via our interactive website; Kidney Interactive Transcriptomics (http://humphreyslab.com/SingleCell/). Source data are provided with this paper.

## Code availability

No customized code was used for data analyses in this study. Analyses were following publicly available instructions from Seurat (http://

satijalab.org/seurat/), Signac (https://satijalab.org/signac/), Cicero (https://cole-trapnell-lab.github.io/cicero-release/docs_m3/), CellChat (http://www.cellchat.org/), CIBERSORTx (https://cibersortx.stanford.edu/), and plot1cell (https://github.com/TheHumphreysLab/plot1cell). The codes used in this study are available on GitHub at https://github.com/TheHumphreysLab/Multimodal_analysis_ADPKD.

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

## Acknowledgements

These experiments were funded by a sponsored research agreement from Chinook Therapeutics and The Baltimore Polycystic Kidney Disease Research and Clinical Core Center (NIDDK, P30DK090868). Additional support was from the Japan Society for the Promotion of Science (JSPS) Postdoctoral Fellowships for Research Abroad and The Osamu Hayaishi Memorial Scholarship for Study Abroad (Y.M.).

## Author contributions

Y.M. and B.D.H. conceived, coordinated, and designed the study. Y.M. performed experiments with contributions from E.E.D., Y.Y., and K.O. Y.M., H.W., and E.O. performed bioinformatic analysis. Y.M., E.E.D., Y.Y., H.W., A.J.K., E.O., M.G., J.K., P.A.W., T.J.W., K.O., J.H.M., N.L., P.C.W., and B.D.H. analyzed data. ADPKD samples were collected by E.E.D., S.L.S., O.M.W., and T.J.W. Y.M. and B.D.H. designed experiments and wrote the manuscript. All authors read and approved the final manuscript.

## Competing interests

B.D.H. is a consultant for Janssen Research & Development, LLC, Pfizer and Chinook Therapeutics, holds equity in Chinook Therapeutics and grant funding from Chinook Therapeutics and Janssen Research & Development, LLC. O.M.W. has received grants from AstraZeneca unrelated to the current work. J.H.M. has received funding from Chinook Therapeutics unrelated to the current work. S.S. has received grant funding from Otsuka, Palladio Biosciences, Kadmon Corporation, Sanofi, and Reata Pharmaceuticals. A.J.K., E.O., M.G., J.K., and J.H.C. are employees and stock holders of Chinook Therapeutics. The remaining authors declare no competing interests.
