## [Peer Review File · Nature Communications]

Reviewers' Comments:

Reviewer #1:

Remarks to the Author:

This study utilizes sn-RNAseq to analyze transcriptional changes in ADPKD kidney tissue. Eight ADPKD and 5 control kidneys were analyzed. In addition, epigenetic studies employed snATAC-seq to identify open chromatin regions. Analysis of proximal tubules showed a failed repair transcriptomic signature characterized by profibrotic and proinflammatory transcripts. In collecting ducts, the orphan G protein-coupled receptor GPRC5A was upregulated, that was associated with increased transcription factor binding motif availability for the NF-KB, TEAD, CREB and retinoic acid receptor families, associated with a distal enhancer associated with GPRC5A. This comprehensive view of ADPKD associated expression changes is likely to be important for better understanding key pathways associated with ADPKD pathogenesis.

The major value of this study is the snRNAseq and snATAC-seq data that will be deposited for other investigators to study. However, the authors have made efforts, through a few vignettes of specific changes with additional staining data to show the significance of the work. These explorations do not go too deeply or are very mechanistic but do show examples of what can be gleaned from the data.

Specific Points:

1. A significant caveat of this work is that the ADPKD material comes from ESKD patients. In contrast, and as indicated, the normal kidney subjects had normal kidney function. This deficiency of the study should be more clearly acknowledged in the publication.
2. More details should be shown of what region of the kidney in the ADPKD patients and normal controls was used for the analysis. What degree of variability was seen between different patients, and could this be associated with the region sampled?
3. Is the genetic cause of the ADPKD known in these patients? Was it possible to determine this information from the sequence data? Were there expression level differences of PKD1 and PKD2 gene between the ADPKD kidneys, and compared to the control kidneys?
4. A considerable amount of RNA seq data from various PKD models, usually at early stages, as well as some data from late-stage human tissue has been published with data deposited in appropriate repositories. It would be helpful to compare and contrast the finding in this study to the data presently available.
5. Information about the role of fibrosis in ADPKD are identified. However, these are likely features of late-stage disease reflecting the starting material, and this should be acknowledged.
6. Control kidney datasets had more unique genes and transcripts per cell than ADPKD samples, likely reflecting increased RNA degradation during dissociation from the more fragile ADPKD tissue (Supplementary Fig. 3). This issue should be discussed.
7. Was the Hedgehog pathway also activated in other renal cell types in addition to fibroblasts in ADPKD kidneys?

Minor points:

1. Since polycystin 1 and 2 and called PC1 and PC2, naming PKD expression subpopulations PKD-PC1 etc. is confusing.

Reviewer #2:

Remarks to the Author:

In this manuscript, Muto et al. present single nuclear RNA-seq and ATAC-seq data in kidneys from patients with autosomal dominant polycystic kidney disease (ADPKD) and controls, respectively. The overall quality of the data seems to be high with respect to sequencing depth, although systematic differences between control and ADPKD samples are noticeable, potentially due to the cystic nature of the ADPKD samples. The analytical pipelines used are standard and state-of-the-art except for trajectory analyses, which in my view present serious issues and should be reproduced after sufficient batch correction and with different software packages including RNA velocity analysis. Analyzing either 102,710 (RNA-seq) or 50,986 nuclei (ATAC-seq) datasets, the authors describe the cellular architecture of ADPKD kidneys and focus the presentation of their results on several cell types. Most importantly, they identify a distal enhancer regulating GPRC5A

in one cell type and validate their findings with CRISPR interference in another. A confusing aspect of this manuscript is the somewhat selective results presentation of the rich dataset, sometimes prioritizing findings that connect two parts of a biologically interesting story over the most obvious and strongest signals in the data. Despite this, the story flow is still confusing, jumping from failed repair proximal tubules to interstitial cells to endothelial cells to collecting duct epithelial cells. What is also striking is the highly speculative nature of some concluding remarks, e.g., regarding autocrine and paracrine functions without any ligand-receptor analysis performed. In my view, substantial revision of which parts of the data are presented and in which order would be necessary to make for a more coherent story. Ligand-receptor analyses should be added. Spatial transcriptomics methods, which have successfully been used in the Humphreys lab before, would be suited to confirm or refute some of the speculative hypotheses pertaining to autocrine and paracrine cell signaling and add a potential aspect of novelty. My detailed comments are as follows:

Major:

- Although no eGFR at the time of kidney transplant is given for sampled ADPKD patients, all patients had end stage kidney disease and substantial kidney weight of the explanted and sampled kidneys (Suppl. Table 1), implying the presence of a typical ADPKD morphology with large, potentially giant cysts. In addition to giving the respective eGFRs for each patient, can the authors please describe in detail their pragmatic approach to sampling strategy given the perceived difficulty of sampling cystic tissues: Was the sampling performed by the researchers or by a surgeon? Was the biopsy taken from tissue "between" large cysts at the curvature of the kidney or from a cut tissue section where the size of cysts was visible a priori? What percentage of the sample was destructed by large, mid-size, small cysts? How did the single nuclei preparation differ from control kidneys, which must have been much more homogenous compared to cystic kidney tissue? Do the authors think the lower quality of ADPKD samples is mostly due to the nuclei preparation process and difficulties pertained with the cystic nature of those kidneys? How would this explain the substantially lower number of genes or UMIs per cell and the lower fraction of reads in peaks, respectively (Suppl. Fig. 3)?

- The authors should give fractions of normal (N-PTC) and failed repair proximal tubule cell (FR-PTC) clusters for ADPKD vs. control clusters. It is impossible to verify from Fig. 3b whether the claim that "most of the PT in ADPKD kidneys were FR-PTC, while most of PT in control were N-PTC" is true. The authors should give fraction bar graphs and a UMAP colored by disease group and by individual samples. What strikes me are the ostensibly even numbers of FR- and N-PTC cells, respectively. In previous studies from the same group (very strong ischemia damage, human diabetic kidney disease samples, etc.) the fractions of FR-PTC were substantially lower (<4%). How do the authors explain this stark overrepresentation of FR-PTC in this particular dataset? Did the authors use all PT cells for subclustering analysis, as suggested from Fig. 3a, or did they subsample even numbers for FR-PTC and N-PTC? Was Harmony really used for batch correction also in the subclustering analysis, as suggested in the Methods for RNA-seq (Harmony is not mentioned for ATAC-seq...)? I am worried the 2 separate and seemingly almost even-numbered clusters could just represent a serious batch effect (left cluster control samples, right cluster ADPKD samples), or possibly sampling effect. It is hard to imagine that nearly all PT cells – even in an ESKD kidney – resemble FR-PTC, as it is known that even in largely destroyed cystic kidneys, there are scattered, albeit small, areas of relatively intact parenchyma and it is hard to believe that there are almost no healthy PT cells left (after all, at least the 1 ADPKD patient with pre-emptive kidney transplant should have contributed a very substantial amount of healthy PT cells!). Along those lines, the authors cite Suppl. Figs. 1a and 2a when describing their VCAM1 cluster, however, I cannot detect a FR-PTC cluster in either of both these figures. Suppl. Fig. 1a does not even show VCAM1 as a marker.

- Along those lines, I suspect the trajectory analysis (Fig. 3d) to be seriously compromised, possibly by batch effect. The authors describe a "gradual reduction" of healthy PT marker expression and increased FR-PTC marker expression, respectively. On the contrary, it is evident from both the UMAP and the gene expression vs. pseudotime scatterplots that there is a sudden and abrupt change rather than continuous transition. There are no smooth connections between the 2 clusters, as evidenced by a gap in the trajectory between N-PTC and FR-PTC, potentially

indicating an artifact due to insufficient batch integration or such as one would get when analyzing a “trajectory” between unrelated cell types. Again, the authors should demonstrate sufficient batch effect correction of ADPKD and control samples in their trajectory analysis. The results should be replicated with other software than monocle3, which is known to be prone to these issues, as it relies on UMAP. I would also like to insist to see the trajectory results compared to RNA velocity analysis using, e.g., scVelo, which is less biased and potentially reveals true biological trajectories more faithfully. Also, the authors should add trajectory analyses of their ATAC-seq dataset to be able to compare consistencies and differences between the two methods.

- TGFB2 chromatin accessibility: The authors suggest autocrine or paracrine effects of TGFbeta signaling related to FR-PTC. It is necessary to show TGFB2 chromatin accessibility in a plot showing all cell types including N-PTC and FR-PTC, split by ADPKD vs. control groups, in order to be able to gauge specificity for FR-PTC or ADPKD-derived PT cells and make a respective statement. It is not clear to me whether all PT cells were used for the chromatin accessibility plot in Fig. 3h (as the figure legend suggests) or just FR-PTC (as the manuscript text implies).

- The immunofluorescence images in Fig. 3c show very high background in the green channel. From these images, it is also impossible to ascertain whether the VCAM1 staining is actually in PT or other tubular cells. Given the strikingly high percentage of FR-PT cells in ADPKD samples according to the scRNA-seq dataset, double stains of VCAM1 with a PT marker with optimized image exposure should be performed to demonstrate that VCAM1 positive cells are indeed PT cells. Alternatively, the authors should use FISH or another technique to demonstrate the presence of VCAM1/VCAM1 in PT cells and cyst-lining cells.

- Interstitial cells and paracrine signaling: The authors suggest potential paracrine signaling of TNFalpha between collecting duct-derived cysts and interstitial ADPKD-specific fibroblasts (PKD-FIB). At the minimum, a ligand-receptor analysis including all cell types should be performed in order to rectify such a suggestion/hypothesis. This type of analysis will also be able to test the authors’ prior claims/hypotheses of autocrine and paracrine signaling of FR-PTC as well as their later hypotheses of paracrine DHH signaling from arterial endothelial cells (AEC) to surrounding pericytes and fibroblasts.

- On a similar note, most of these speculative notions (which cell types might interact with one another in a paracrine fashion, as suggested both by their spatial proximity and their expression pattern of ligands and receptors) could be approached using spatial transcriptomics methods, which are available and have been used previously in the Humphreys lab. In this way, the proximity of FR-PTC, endothelial, interstitial, and other epithelial cells such as the collecting duct principal cells could be elucidated.

- It is very confusing why the authors highlight GLI1 expression (which is hardly present in more than 2% of Myofib or PKD-FIB cells) rather than the more obvious finding from their dataset (IL6 expression in PKD-FIB, which was highly expressed in possibly 40-50% of cells, gauging from Suppl. Fig. 6b). The statement “GLI1 was highly expressed in FIB of ADPKD kidneys” seems overly optimistic to say the least, given the scarce positivity of cells in both Myo-FIB and PKD-FIB clusters (2% at the most!). The fact that the authors show the UMAP of all cells (Fig. 4j) rather than the subclustering UMAP for GLI1 expression (Suppl. Fig. 6d) in the main figure is misleading in this context. The dot plot visualizing avg. expression level and percentage of cells expressing the marker (Fig. 4g) should include all interstitial cells (including unknown1 and unknown2). From this plot, it is hard to gauge the fraction of cells expressing GLI1, it might make sense to produce a separate dot plot because it looks like only ~1-2% of PKD-FIB cells are actually expressing GLI1. Other than for the story connecting fibroblasts with Hedgehog pathway, it does not occur to me why the authors highlight this part of their data rather than the more striking and interesting finding from the subclustering analysis (IL6 expression in PKD-FIB cells). Most concerning to me is the fact that the low percentage of cells expressing GLI1 was not acknowledged in the manuscript text.

- Collecting duct cyst subtypes: Again, it looks as if the subclustering analysis has a substantial remaining batch effect, as evidenced by the substantial overlap between control and ADPKD cells before and hardly any overlap after subclustering, respectively (Fig. 5a). This might also explain

the somewhat problematic trajectories with major discontinuities and noticeable gaps, again to be verified with other software packages and complemented by RNA velocity analysis.

- In the discussion, the authors suggest that "differences in responses to hypoxia may switch the trajectories to PKD-PC1 and PKD-PC2", which seems to me very hypothetical, as the authors do not show any analyses in hypoxia and ischemia datasets that would indicate so. Again, the trajectories towards PKD-PC1 and PC2, respectively, seem problematic because of the substantial discontinuities, as outlined above.

- The authors report differential accessibility for a distal enhancer of GPRC5A in seemingly PKD-specific collecting duct principal cells (PCs) and validate this by CRISPR interference not in PCs, but in proximal tubule cells (PTC), which is in itself strange and incoherent. Can the authors verify the specificity of the CCAN for PCs? They should show corresponding gene browser views visualizing chromatin accessibility for all cell types (PCT, PST, FR-PTC, PEC_PODO, TAL, DCT, CNT_PC, IC, ENDO, FIB, LEUK) with a corresponding CCAN. I am concerned that their results pertaining to chromatin accessibility of GPRC5A and its distal enhancer are not specific to collecting duct but also present in other cell types, namely PT cells. Also, have the authors performed similar CRISPR interference assays for the distal enhancer element of MIR31HG?

Minor:

- What do the authors mean exactly when they say that increased promoter accessibility of ADPKD compared to control samples "... suggest[s] dynamic epigenetic remodeling"? This is somewhat of an imprecise statement. How do the authors explain this systematically higher promoter accessibility of ADPKD samples compared to control samples, which was true across all cell types?

Reviewer #1 (Remarks to the Author):

This study utilizes sn-RNaseq to analyze transcriptional changes in ADPKD kidney tissue. Eight ADPKD and 5 control kidneys were analyzed. In addition, epigenetic studies employed snATAC-seq to identify open chromatin regions. Analysis of proximal tubules showed a failed repair transcriptomic signature characterized by profibrotic and proinflammatory transcripts. In collecting ducts, the orphan G protein-coupled receptor GPRC5A was upregulated, that was associated with increased transcription factor binding motif availability for the NF-KB, TEAD, CREB and retinoic acid receptor families, associated with a distal enhancer associated with GPRC5A. This comprehensive view of ADPKD associated expression changes is likely to be important for better understanding key pathways associated with ADPKD pathogenesis.

The major value of this study is the snRNaseq and snATAC-seq data that will be deposited for other investigators to study. However, the authors have made efforts, through a few vignettes of specific changes with additional staining data to show the significance of the work. These explorations do not go too deeply or are very mechanistic but do show examples of what can be gleaned from the data.

[Response] We thank the reviewer for the careful evaluation of our manuscript. Our responses to the specific points raised follow:

Specific Points:

1. A significant caveat of this work is that the ADPKD material comes from ESKD patients. In contrast, and as indicated, the normal kidney subjects had normal kidney function. This deficiency of the study should be more clearly acknowledged in the publication.

[Response] We agree with this point. Unfortunately endstage ADPKD human material is the only type available. We now more clearly acknowledge this limitation in the discussion (**page 11, line 41 - 44**).

2. More details should be shown of what region of the kidney in the ADPKD patients and normal controls was used for the analysis. What degree of variability was seen between different patients, and could this be associated with the region sampled?

[Response] The ADPKD samples investigated here were collected from the base (cup) of large cortical, superficial cysts (see images of all ADPKD kidneys below and now added to the Supplementary Fig. 1. Control samples were obtained from the outer cortex of non-tumor kidney tissue nephrectomized from patients with normal kidney function (**New Supplementary Fig. 1**). We also added this information in Results and Method section (**page 4, line 4 - 5; page 13, line 10 - 12**)

New Supplementary Figure 1. Gross appearance of ADPKD kidney samples used for single cell analysis: ADPKD kidneys nephrectomized from ESKD patients. An asterisk (*) indicates the cyst from which a sample for single cell analysis was collected in each patient.

We observed considerable variability in celltype frequency in ADPKD samples compared to controls (**New Supplementary Fig. 5**). This variability may reflect the location of the cyst that was sampled (cortical, corticomedullary or medullary region). We have included this new analysis in the supplementary materials and comment in the text (**page 4, line 30-33**).

New Supplementary Figure 5. Proportion of cell lineages in each dataset
 The proportion of cell lineages in dataset from each sample. Proximal, PT, FR-PTC, PEC and PODO; Distal, TAL1, TAL2, DCT, CNT_PC and IC.

3. Is the genetic cause of the ADPKD known in these patients? Was it possible to determine this information from the sequence data? Were there expression level differences of *PKD1* and *PKD2* gene between the ADPKD kidneys, and compared to the control kidneys?

[Response] Unfortunately the genetic cause of ADPKD was not known in these patients. We attempted to identify mutations based on mapped reads on our snRNA-seq dataset, but this was unsuccessful likely due to low overall expression and the 3' bias of our libraries. We compared *PKD1* and *PKD2* mRNA expression in healthy vs. ADPKD samples. If anything, expression of these genes was higher in ADPKD (**New Supplementary Fig. 6; page 4, line 33-36**).

New Supplementary Figure 6. *PKD1* or *PKD2* gene expressions in ADPKD dataset.

(a) UMAP plot displaying *PKD1* (upper panel) or *PKD2* (lower panel) gene expression in control or ADPKD dataset. The color scale for each plot represents a normalized log-fold-change (LFC). (b) Dot plot showing *PKD1* (left) or *PKD2* (right) gene expression in each celltype of control or ADPKD dataset.

4. A considerable amount of RNA seq data from various PKD models, usually at early stages, as well as some data from late-stage human tissue has been published with data deposited in appropriate repositories. It would be helpful to compare and contrast the finding in this study to the data presently available.

[Response] To address this comment, we performed deconvolution analysis on published microarray data of human ADPKD kidneys (GSE7869) with our dataset using CIBERSORTx, machine learning method that imputes gene expression profiles and estimates the frequency of cell types in a mixed cell population (Nat. Biotechnol., 2019, Jul;37(7):773-782).

New Figure 6. Deconvolution analysis of human ADPKD cyst dataset

(e) Predicted frequencies of celltypes in each dataset of normal kidney cortex (n=3) of healthy control, and minimal cystic tissue (n=5) or renal cyst (n=13) of ADPKD patients in deconvolution analysis of human ADPKD kidney datasets (GSE7869). The predicted FIB frequencies in each group are also shown (right). (f) Predicted relative gene expressions of *PDGFRB*, *ACTA2*, *FN1*, *IL6* or *FGF14* in FIB of each group. Bar graphs represent the mean and error bars are the s.d. One-way ANOVA with post hoc Dunnett's multiple comparisons test.

This deconvolution analysis predicted a significant increase in the fibroblasts (FIB) population in cystic kidneys compared to either minimally cystic or normal kidneys (New Fig. 6e). Celltype-specific expression purification at high resolution with CIBERSORTx predicted up-regulation of Myo-FIB markers (*ACTA2*, *FN1*) and PKD-FIB markers (*IL6* and *FGF14*, New Fig. 6f), suggesting expansion of these FIB subsets. These results suggest that PKD-FIB and Myo-FIB subsets are associated with large cysts in ADPKD kidneys. We have now included these points in the revised manuscript (page 7, line 13 - 20).

We also applied our dataset to mouse ADPKD model dataset (GSE86507) with CIBERSORTx (Fig. R1, for reviewing purposes only). In contrast to the human microarray dataset, CIBERSORTx did not predict any significant changes in celltype frequencies, possibly due to discrepancy between human single-nucleus transcriptomic profile and mouse bulk (mainly cytoplasmic) transcriptomic profile.

Figure R1. Deconvolution analysis of mouse ADPKD cyst dataset

(a) Predicted frequencies of celltypes in each dataset for kidneys from ADPKD model mice with collecting-duct-specific inactivation of either *Pkd1* or *Pkd2* gene (*Pkd1*^{fl/fl}: HoxB7-Cre or *Pkd2*^{fl/fl}: HoxB7-Cre) kidneys and their controls at postnatal day (P)1, P3 or P7 (GSE86507). (b) The predicted FIB frequencies in each group are shown. Bar graphs represent the mean and error bars are the s.d. One-way ANOVA with post hoc Dunnett’s multiple comparisons test.

5. Information about the role of fibrosis in ADPKD are identified. However, these are likely features of late-stage disease reflecting the starting material, and this should be acknowledged.

[Response] We agree with this point. We have now acknowledged this point in the revised manuscript (page 10, line 37-39).

6. Control kidney datasets had more unique genes and transcripts per cell than ADPKD samples, likely reflecting increased RNA degradation during dissociation from the more fragile ADPKD tissue (Supplementary Fig. 3). This issue should be discussed.

[Response] We agree. We now more specifically call this out in the revision (page 4, line 19-25).

7. Was the Hedgehog pathway also activated in other renal cell types in addition to fibroblasts in ADPKD kidneys?

[Response] GLI1 is known to be an amplifier of hedgehog signaling (HH) pathway, induced by GLI2/GLI3 that respond to initial HH activation. *GLI1* transcripts were detected mainly in FIB cluster, and PEC to a less extent (Fig. R2a). In contrast, single-cell enrichment analysis of a gene set "HALLMARK_HEDGEHOG_SIGNALING" (from MSigDB, Broad Institute), that consists of genes up-regulated by activation of hedgehog signaling (Fig. R2b, New Fig. 3a), suggested that HH pathway was specifically activated in podocyte (PODO) cluster both in ADPKD and control datasets. The discrepancy between GLI1 expression pattern and the gene set enrichment analysis result may be due to inadequate detection of genes related to HH signaling pathways in snRNA-seq on ADPKD kidneys. Furthermore, ligand-receptor analysis predicted that HH signaling originated from PT and TAL, and the predicted targets were also PT and TAL in ADPKD kidneys (Please also see the response to the comment 2-6). These findings were not consistent with the notion in the original manuscript that HH signal might be from an endothelial subset to fibroblasts. We decided to remove the hypothesis from the manuscript, since our data did not supply enough evidence to suggest this notion. Future spatial transcriptomic analysis on ADPKD samples may be able to test this hypothesis in the future (Please also see the response to the comment 2-7).

Figure R2. Hedgehog signaling in ADPKD kidney dataset

(a) UMAP (left) or violin plot (right) displaying *GLI1* gene expression.

(b) UMAP (left) or violin plot (right) displaying gene set enrichment analysis (GSEA) of a gene set "HALLMARK_HEDGEHOG_SIGNALING" (genes up-regulated by activation of hedgehog signaling) at a single-cell resolution.

Minor points:

1. Since polycystin 1 and 2 are called PC1 and PC2, naming PKD expression subpopulations PKD-PC1 etc. is confusing.

[Response] We agree with the reviewer that naming the PKD-specific principal cell subsets PKD-PC1/PKD-PC2 is confusing. We changed their names to PKD-CDC1/2 (PKD-specific collecting duct cell 1 or 2 subset) throughout the manuscript.

Reviewer #2 (Remarks to the Author):

In this manuscript, Muto et al. present single nuclear RNA-seq and ATAC-seq data in kidneys from patients with autosomal dominant polycystic kidney disease (ADPKD) and controls, respectively. The overall quality of the data seems to be high with respect to sequencing depth, although systematic differences between control and ADPKD samples are noticeable, potentially due to the cystic nature of the ADPKD samples. The analytical pipelines used are standard and state-of-the-art except for trajectory analyses, which in my view present serious issues and should be reproduced after sufficient batch correction and with different software packages including RNA velocity analysis. Analyzing either 102,710 (RNA-seq) or 50,986 nuclei (ATAC-seq) datasets, the authors describe the cellular architecture of ADPKD kidneys and focus the presentation of their results on several cell types. Most importantly, they identify a distal enhancer regulating GPRC5A in one cell type and validate their findings with CRISPR interference in another. A confusing aspect of this manuscript is the somewhat selective results presentation of the rich dataset, sometimes prioritizing findings that connect two parts of a biologically interesting story over the most obvious and strongest signals in the data. Despite this, the story flow is still confusing, jumping from failed repair proximal tubules to interstitial cells to endothelial cells to collecting duct epithelial cells. What is also striking is the highly speculative nature of some concluding remarks, e.g., regarding autocrine and paracrine functions without any ligand-receptor analysis performed. In my view, substantial revision of which parts of the data are presented and in which order would be necessary to make for a more coherent story. Ligand-receptor analyses should be added. Spatial transcriptomics methods, which have successfully been used in the Humphreys lab before, would be suited to confirm or refute some of the speculative hypotheses pertaining to autocrine and paracrine cell signaling and add a potential aspect of novelty. My detailed comments are as follows:

[Response] We thank the reviewer for the careful evaluation of our manuscript and constructive suggestions. We agree that trajectory inference had problems, based on the additional analyses with different approaches (i.e. RNA velocity / Monocle 2). We have substantially revised the original Fig.3 (analyses on proximal tubular cell cluster) and removed trajectory inference because of the reasons detailed below. We performed CRISPRi on the distal enhancers for *GPRC5A* and *MIR31HG* using an ADPKD cyst cell line, and we removed the data with primary RPTEC to avoid confusion. We also appreciate the suggestion that "*revision of which parts of the data are presented and in which order would be necessary to make for a more coherent story*". To make the story flow more coherent, we added new Fig. 3 and Fig. 4 to show why we focused on proximal tubular cells, interstitial cells and collecting duct epithelial cells.

We observed that various inflammatory pathways (IL6-mediated STAT3 activation and NF-kB activation) as well as TGF β signaling pathway were generally activated in ADPKD microenvironment in single cell gene set enrichment analysis (**New Fig. 3a, b**) in snRNA-seq, validated by motif enrichment analysis of the transcription factors in snATAC-seq (**New Fig. 3c-e; page5, line 10 - 20**).

New Figure 3. Activation of inflammatory, profibrotic pathways in ADPKD kidneys
(a) Heatmap showing enrichment of hallmark gene sets of the Molecular Signatures Database (MsigDB) in each cell type of ADPKD or control kidneys. **(b)** UMAP displaying enrichment of genes regulated by NF- κ B pathway in response to TNF α (upper), genes up-regulated by IL6 via STAT3 (middle) or genes up-regulated in response to TGF β signaling (lower) in snRNA-seq dataset. **(c-e)** UMAP displaying enrichment of transcription factor binding motifs in control or ADPKD kidneys (left) or violin plot showing the relative motif enrichment scores in each cell type (right) for RELA **(c)**, STAT3 **(d)** or SMAD2/SMAD3/SMAD4 complex **(e)**. The color scale represents a normalized log-fold-change (LFC).

We performed ligand-receptor analyses with CellChat to quantitatively infer cell-cell communication networks. This analysis identified three primary cell types and the ligand they predominately secreted: IL6 by fibroblasts, TNF by collecting duct epithelial cells and TGF β by proximal tubular cells. New Figure 4 shows this analysis, including target cells responding to these three ligands and the signal receiver celltypes. (New Fig. 4, page5, line 20 - 24).

New Figure 4. Ligand-receptor analysis identified proinflammatory, profibrotic signaling network: (a) Dot plot showing gene expression of *TNF* (upper), *IL6* (middle) or *TGFB2* (lower) in each cell type in ADPKD or control kidneys. The diameter of the dot corresponds to the proportion of cells expressing the indicated gene and the density of the dot corresponds to average expression relative to all cell types. (b) Ligand-receptor analysis with CellChat. Circle plot showing an inferred network (left) or heat map showing communication probabilities from senders (secretors) to receivers (targets, right) for TNF signaling pathway (upper), IL6 signaling pathway (middle) or TGF β signaling pathway (lower). Thickness of an arrow in a circle plot indicates interaction strength.

Based on these results, we revised the story to center on the inflammatory/profibrotic signaling pathways activated by these three cell types. We removed the analyses on endothelial cell cluster and hedgehog signaling pathway to avoid confusion and distraction. We tried Visium spatial transcriptomics on human kidney, but the pilot failed (only the highest expressed genes were detected). We believe this is the consequence of inadequate permeabilization. Given the very high cost of these studies, and the need to re-optimize a workflow for human samples and using formalin-fixed ADPKD samples (the only ones available to us), we respectfully submit that these studies are beyond the scope of the current manuscript.

Our responses to the specific points raised follow:

Major:

I- Although no eGFR at the time of kidney transplant is given for sampled ADPKD patients, all patients had end stage kidney disease and substantial kidney weight of the explanted and sampled kidneys (Suppl. Table 1), implying the presence of a typical ADPKD morphology with large, potentially giant cysts. In addition to giving the respective eGFRs for each patient, can the authors please describe in detail their pragmatic approach to sampling strategy given the perceived difficulty of sampling cystic tissues: Was the sampling performed by the researchers or by a surgeon? Was the biopsy taken from tissue “between” large cysts at the curvature of the kidney or from a cut tissue section where the size of cysts was visible a priori? What percentage of the sample was destructed by large, mid-size, small cysts? How did the single nuclei preparation differ from control kidneys, which must have been much more homogenous compared to cystic kidney tissue? Do the authors think the lower quality of ADPKD samples is mostly due to the nuclei preparation process and difficulties pertained with the cystic nature of those kidneys? How would this explain the substantially lower number of genes or UMIs per cell and the lower fraction of reads in peaks, respectively (Suppl. Fig. 3)?

[Response] We agree with reviewer 2 and we now provide more detail on these samples, see below.

IA. - Although no eGFR at the time of kidney transplant is given for sampled ADPKD patients, all patients had end stage kidney disease and substantial kidney weight of the explanted and sampled kidneys (Suppl. Table 1), implying the presence of a typical ADPKD morphology with large, potentially giant cysts. In addition to giving the respective eGFRs for each patient, can the authors please describe in detail their pragmatic approach to sampling strategy given the perceived difficulty of sampling cystic tissues: Was the sampling performed by the researchers or by a surgeon? Was the biopsy taken from tissue “between” large cysts at the curvature of the kidney or from a cut tissue section where the size of cysts was visible a priori? What percentage of the sample was destructed by large, mid-size, small cysts?

All patients had end-stage kidney disease with eGFR < 20 ml/min/1.73m² as required for transplantation eligibility. We added this information (eGFR <20 ml/min/1.73m²) to the patient information table (**New Supplementary Table 1**) for clarity. The Maryland PKD Research and Translation Core Center, in a unique relationship with a team of transplant surgeons at the University of Maryland Medical Center, has developed a protocol for the PKD center to receive intact nephrectomized kidneys as soon as they are removed from the transplant patient, in the operation room. The PKD researcher then immediately begins the dissection and collection of samples. The kidneys are cooled on ice as soon as they are removed from the patient and during the entire dissection and sample preparation process. The samples investigated here were collected from the base (cup) of large cortical / superficial cysts (see images now added to the **New Supplementary Fig. 1**) with portions fixed and portions flash frozen. By design, each sample contained the epithelial wall of one large cyst, but previous analysis demonstrated that the samples also contained significant other cysts of all sizes (data not shown). We are not able to quantitate percentage of various cyst sizes since each sample is unique. We have now added these points in Method section (**page 13, line 3-14**).

1B. - How did the single nuclei preparation differ from control kidneys, which must have been much more homogenous compared to cystic kidney tissue? Do the authors think the lower quality of ADPKD samples is mostly due to the nuclei preparation process and difficulties pertained with the cystic nature of those kidneys? How would this explain the substantially lower number of genes or UMIs per cell and the lower fraction of reads in peaks, respectively (Suppl. Fig. 3)?

We performed single nuclei preparation on ADPKD tissues using the exact same protocol as we did on control kidneys. We did observe more debris in nuclei suspensions from ADPKD kidneys probably due to more fibrotic and necrotic nature of late stage CKD samples. Fibrotic tissue increases friction during mechanistic dissociation of tissue/nuclei using Dounce homogenizer, so more nuclei may be damaged resulting in the observed lower quality libraries from ADPKD samples. We have now added this point in our manuscript (**page 4, line 20-25**).

2A. - The authors should give fractions of normal (N-PTC) and failed repair proximal tubule cell (FR-PTC) clusters for ADPKD vs. control clusters. It is impossible to verify from Fig. 3b whether the claim that “most of the PT in ADPKD kidneys were FR-PTC, while most of PT in control were N-PTC” is true. The authors should give fraction bar graphs and a UMAP colored by disease group and by individual samples. What strikes me are the ostensibly even numbers of FR- and N-PTC cells, respectively.

[Response] We have quantitated the fraction of FR-PTC among PT lineage of each sample and now present the results as a bar graph (**New Fig. 5c**). The fractions of FR-PTC for control PTC (9.2 +/- 4.3%) were similar to our previous analysis (GSE151302, 7.9 +/- 3.7%). In contrast, those for ADPKD were significantly higher than controls (98.5 +/- 2.9%). We also presented a UMAP colored by disease group and by individual samples, along with each UMAP for individual samples for better visibility (**New Fig. 5a, b**).

New Figure 5. (a) UMAP of PT lineage colored by subclusters (subtypes, left) or disease group (right). **(b)** UMAP split by individual samples, colored by subclusters.

2B. - In previous studies from the same group (very strong ischemia damage, human diabetic kidney disease samples, etc.) the fractions of FR-PTC were substantially lower (<4%). How do the authors explain this stark overrepresentation of FR-PTC in this particular dataset?

[Response] The mouse model of ischemic damage in our previous study was bilateral ischemia reperfusion injury, and the acute kidney injury was finally resolved in 6 weeks, with slight increase of serum creatinine concentration left (Proc. Natl. Acad. Sci. U. S. A., 2020 Jul 7;117(27):15874-15883). Another study from our group estimated the fraction of FR-PTC in human diabetic nephropathy (DN) samples with deconvolution of published RNA-seq data. The DKD patients in that study had relatively maintained renal function (Early DN: eGFR = 117.7 +/- 8.624 ml/min, Advanced DN: 63.79 +/- 5.765 ml/min) [Diabetes, 2019, Dec;68(12):2301-2314]. In contrast, the ADPKD patients in our study had end stage kidney disease (eGFR < 20 ml/min or already on dialysis). Fibrotic, deteriorated kidney tissues with tubular atrophy in end stage kidney disease may account for the substantially higher fraction of FR-PTC in the PT lineage that we observed. There may also be ongoing ischemia and epithelial injury in these endstage kidneys, also driving the FR-PTC phenotype. We have now added discussion of this point in our manuscript (**page 5, line 35-38**).

2C -Did the authors use all PT cells for subclustering analysis, as suggested from Fig. 3a, or did they subsample even numbers for FR-PTC and N-PTC? Was Harmony really used for batch correction also in the subclustering analysis, as suggested in the Methods for RNA-seq (Harmony is not mentioned for ATAC-seq...)?

[Response] We used all PT cells for subclustering, and we did not subsample, and we also used Harmony with an argument: group.by.vars = "orig.ident" (sample) to correct for batch effects among samples in the original manuscript as described in the Methods (**page 15, line 37-38**). We apologize for missing the detail of batch correction in snATAC-seq subclustering. Subclustering in snATAC-seq dataset was performed with the use of Harmony embedded on the whole dataset. We added these details onto the revised method section (**page 15, line 42-43**).

2D -I am worried the 2 separate and seemingly almost even-numbered clusters could just represent a serious batch effect (left cluster control samples, right cluster ADPKD samples), or possibly sampling effect.

[Response] Both ADPKD and control PT lineage consisted of N-PTC and FR-PTC subsets, although the proportion of these subtypes was significantly skewed in each disease group (i.e. most of control cells were N-PTC, and most of ADPKD cells were FR-PTC). Similar gene expression signature was observed between control and ADPKD cells in each subset (N-PTC or FR-PTC, **New Fig. 5d**), suggesting the subclustering successfully classified the PT cells by cell states with inherent molecular signatures, rather than by disease groups with batch difference. We cannot eliminate the possibility that some residual batch

New Figure 5 (d) Dot plot showing expression of marker genes in each disease group (control and ADPKD) of PT subtypes (N-PTC and FR-PTC).

effects remain due to general lower QC of ADPKD dataset than control, despite our use of Harmony and batch removal of low QC cells in the dataset preprocessing. We have now addressed these points and acknowledged the possibility of batch effect remaining in PT subclustering as limitations in the revised manuscript (**page 5, line 40-45**).

2E -It is hard to imagine that nearly all PT cells – even in an ESKD kidney – resemble FR-PTC, as it is known that even in largely destroyed cystic kidneys, there are scattered, albeit small, areas of relatively intact parenchyma and it is hard to believe that there are almost no healthy PT cells left (after all, at least the 1 ADPKD patient with pre-emptive kidney transplant should have contributed a very substantial amount of healthy PT cells!).

[Response] Our ADPKD datasets include 7 ADPKD patients with pre-emptive kidney transplantation and 1 patient on maintenance hemodialysis. All of the patients had significant renal dysfunction (eGFR < 20 ml/min). We co-stained VCAM1 along with CDH6, which is a proximal nephron marker (**New Fig. 5e**, please also see the below response to the reviewer comment 5), on the ADPKD kidney sections. We observed a large part of CDH6+ tubular cells were VCAM1+ (**New Fig. 5f**), although we also observed CDH6+VCAM1- tubules that look normal in the same samples (**New Fig. 5f**) as suggested by the reviewer. Significant heterogeneity of the frequencies of VCAM1+ cells in CDH6+ tubules among samples and among areas of even the same sample make quantification very difficult. It is possible that our nuclear dissociation biased towards representation of FR-PTC in the ADPKD kidneys, although we have no direct evidence to suggest this. We have now added discussion of this point as a limitation in our manuscript (**page 6, line 15-22**).

New Figure 5 (e) Immunohistochemistry analysis on human healthy adult kidney for CDH6 protein, from human protein atlas (left). Immunofluorescence analysis on human healthy adult kidney for CDH6 (red) costained with CDH1 (green). **(f)** Immunofluorescence analysis on serial sections of ADPKD kidney for VCAM1 (red, left) or CDH6 (purple, right) costained with CDH1 (green). Each pair of white numbers shared between left and right images indicates an identical cyst. VCAM1+CDH6+CDH- tubules/cyst indicate FR-PTC (upper; 3, 4) and VCAM1-CDH6+CDH- tubules indicate N-PTC (lower, 7).

2F - Along those lines, the authors cite Suppl. Figs. 1a and 2a when describing their VCAM1 cluster, however, I cannot detect a FR-PTC cluster in either of both these figures. Suppl. Fig. 1a does not even show VCAM1 as a marker.

[Response] We apologize for this mistake. We meant to cite Supplementary Fig. 1b and 2b (after doublet and low-quality cell filtration) instead of Supplementary Fig. 1a and 2a (before doublet and low-quality cell filtration) in the original manuscript. The PT2 cluster in Supplementary Fig. 1b was FR-PTC, which specifically expressed *VCAMI*. In Supplementary Fig. 2b, we could not distinguish PT and FR-PTC probably due to the low fraction of normal PT in ADPKD kidney dataset (**page 5, line 27-28**). When we integrated control and ADPKD dataset and performed subclustering, N-PT and FR-PTC were distinguished (**New Fig. 5a**).

3A - Along those lines, I suspect the trajectory analysis (Fig. 3d) to be seriously compromised, possibly by batch effect. The authors describe a “gradual reduction” of healthy PT marker expression and increased FR-PTC marker expression, respectively. On the contrary, it is evident from both the UMAP and the gene expression vs. pseudotime scatterplots that there is a sudden and abrupt change rather than continuous transition. There are no smooth connections between the 2 clusters, as evidenced by a gap in the trajectory between N-PTC and FR-PTC, potentially indicating an artifact due to insufficient batch integration or such as one would get when analyzing a “trajectory” between unrelated cell types. Again, the authors should demonstrate sufficient batch effect correction of ADPKD and control samples in their trajectory analysis. The results should be replicated with other software than monocle3, which is known to be prone to these issues, as it relies on UMAP. I would also like to insist to see the trajectory results compared to RNA velocity analysis using, e.g., scVelo, which is less biased and potentially reveals true biological trajectories more faithfully.

[Response] In the pseudotime analysis performed on PT lineage (original Fig. 3d) and CNT_PC (original Fig. 6d), the data were preprocessed for batch correction among samples using R package batchelor [align_cds(cds, num_dim = 10, alignment_group = "orig.ident")]. However, there is still a gap between the two PT subtypes, as reviewer 2 indicates. As suggested, we repeated trajectory analysis with Monocle 2, which relies on UMAP less than Monocle 3. The trajectory generated by Monocle 2 was more continuous, although there was still a sparsity of the cells on the trajectory between N-PTC and FR-PTC (**Fig. R3, for reviewing purposes only**). These findings potentially suggest that two subtypes may be on the steady state without intermediate status (without ongoing transitional process) at the time of sampling from advanced ADPKD kidneys.

Figure R3. Trajectory inference of PT lineage in Monocle 2

(a) Pseudotemporal trajectory from N-PTC to FR-PTC was inferred with Monocle 2 with UMAP colored by subtypes (left) or pseudotime (right). (b) Gene expression dynamics along the pseudotemporal trajectory from N-PTC to FR-PTC are shown (right); *LRP2* (upper left), *SLC5A12* (upper right), *TPM1* (lower left) and *VCAMI* (lower right).

We also performed trajectory inference with RNA velocity (scVero) on PT lineage in snATAC-seq data. The direction of differentiation in each subtype was opposite, suggesting discontinuity of trajectory between N-PTC and FR-PTC (**Fig. R4**).

Figure R4. Trajectory inference of PT lineage with scVero: Trajectory inference was performed with scVero. UMAP was colored by subtypes. The arrows on UMAP indicate the direction of trajectories, and the arrows on each subset were toward opposite direction.

These results are not straightforward. RNA velocity is known to generate arbitrary erroneous directions when cells are mostly in mature states, without intermediate states (Mol. Syst. Biol., 2021, Aug;17(8):e10282). Since ADPKD is chronic kidney disease and takes decades to progress to ESKD, it is possible that there may be far fewer intermediate states compared to a more rapid disease time course like acute kidney injury nephrogenesis. This may be also the reason why we observed a gap on the trajectory between N-PTC and FR-PTC in Monocle 2 and Monocle 3. We are also concerned that applying RNA-velocity to snRNA-seq (rather than scRNA-seq) may be biased because we completely lack most mature mRNAs that are exported into the cytoplasm. This point has been made by others (Mol. Syst. Biol., 2021, Aug;17(8):e10282). Given all of this, we have elected to remove the trajectory analysis from then manuscript.

3B - Also, the authors should add trajectory analyses of their ATAC-seq dataset to be able to compare consistencies and differences between the two methods.

[Response] We performed trajectory analyses on PCT in snATAC-seq with Cicero (batch effect among samples was corrected with batchelor, **Fig. R5**). The trajectory inferred by Cicero was similar to that generated by Monocle 3 in snRNA-seq, leaving a gap between normal PTC and FR-PTC as the snRNA-seq result. As we discussed above, trajectory analysis does not appear to be well suited to apply to our ADPKD dataset.

Figure R5. Trajectory inference of PT lineage in snATAC-seq with Cicero: Pseudotemporal trajectory from N-PTC to FR-PTC using snATAC-seq was generated by Cicero with UMAP colored by subtypes (left) or pseudotime (right).

4- *TGFB2* chromatin accessibility: The authors suggest autocrine or paracrine effects of TGFbeta signaling related to FR-PTC. It is necessary to show *TGFB2* chromatin accessibility in a plot showing all cell types including N-PTC and FR-PTC, split by ADPKD vs. control groups, in order to be able to gauge specificity for FR-PTC or ADPKD-derived PT cells and make a respective statement. It is not clear to me whether all PT cells were used for the chromatin accessibility plot in Fig. 3h (as the figure legend suggests) or just FR-PTC (as the manuscript text implies).

[Response] We now generate coverage plots for all cell types including N-PTC and FR-PTC, split by ADPKD and control (**Fig. R6**). The TSS of *TGFB2* was more accessible in FR-PTC than N-PTC in control kidneys, but was similar between FR-PTC and N-PTC in ADPKD dataset. We also observed accessibility around this TSS in other cell types. This finding is consistent with that *TGFB2* expressions were detected among non-PT celltypes, although the expression level was higher in PT lineage (**New Fig. 4c**). Less specificity of chromatin accessibility around TSS of *TGFB2* may suggest other regulatory mechanisms of *TGFB2* expression besides chromatin accessibility around the TSS. Original Fig. 3h showed all PT cells (We apologize for the misleading manuscript text you mentioned). The difference of accessibility between PT lineages of ADPKD and control cells in original Fig. 3h may be reflected by reduced accessibility of control N-PTC which predominate in the control dataset. To respond the reviewer's suggestion that substantial revision of which parts of the data would be necessary to make for a more coherent story, we removed chromatin accessibility data on TSS of *TGFB2*, since this data does not supply enough clues to understand regulatory mechanisms of *TGFB2* expression in ADPKD kidneys.

5- The immunofluorescence images in Fig. 3c show very high background in the green channel. From these images, it is also impossible to ascertain whether the VCAM1 staining is actually in PT or other tubular cells. Given the strikingly high percentage of FR-PT cells in ADPKD samples according to the scRNA-seq dataset, double stains of VCAM1 with a PT marker with optimized image exposure should be performed to demonstrate that VCAM1 positive cells are indeed PT cells. Alternatively, the authors should use FISH or another technique to demonstrate the presence of VCAM1/VCAM1 in PT cells and cyst-lining cells.

[Response] We thank the reviewer for this suggestion. LTL (Lotus tetragonolobus lectin, green) was only detected in few tubular cells (data not shown) on ADPKD sections probably due to chronic tubular damage. *CDH6* was previously identified as a marker for proximal nephron progenitor cells (J Am Soc Nephrol. 2020 Nov;31(11):2543-2558), and it has been shown to express mainly descending and ascending limb of Henle loop in matured mouse kidneys (Development, 1998, Mar;125(5):803-12). We also observed consistent findings in our mouse kidney dataset (J. Am. Soc. Nephrol., 2019, Jan;30(1):23-32, <http://humphreyslab.com/SingleCell/>). In contrast, we found *CDH6* was expressed in PT/FR-PTC clusters both in ADPKD and control human kidneys in our dataset (**New Supplementary Fig. 10**), although *CDH6* expression was higher in FR-PTC compared to N-PTC. We confirmed that *CDH6* was mainly expressed in proximal nephron in our previously published dataset for human kidneys (Nat. Commun., 2021 Apr 13;12(1):2190). *CDH6* protein was shown to be stained on PT cells in the human kidney in open database (human protein atlas, **New Fig. 5e, left**), and we also confirmed *CDH6* was stained on the *CDH1*-negative non-distal nephron tubular cells (**New Fig. 5e, right**), suggesting that *CDH6* is a proximal nephron marker in adult human kidneys (**page 6, line 4-13**).

New Supplementary Figure 10.
***CDH6* expresses PT lineage in human adult kidneys: UMAP displaying *CDH6* expression in control (left) or ADPKD kidney dataset (right).**

We observed a large part of *CDH6*⁺ tubular cells were *VCAM1*⁺ (**New Fig. 5f, upper panels**), although there were still *CDH6*⁺*VCAM1*⁻ tubules that did not look atrophic in the same samples (**New Fig. 5f, lower panels**) as suggested by the reviewer. Significant heterogeneity of the frequencies of *VCAM1*⁺ cells in *CDH6*⁺ tubules among samples and areas of even the same sample was observed, and it makes quantification quite hard, although observation of non-atrophic *VCAM1*-negative *CDH6*⁺ tubules may not consistent with the frequency of FR-PTC in our dataset. Quantification of a celltype in single nucleus dataset can be influenced by a bias due to tissue sampling, tissue dissociation or nuclei preparation. Such bias may cause overrepresented frequency of FR-PTC in ADPKD dataset. We have added discussion of this point as limitation in our manuscript (**page 6, line 15-22**).

New Figure 5 (e) Immunohistochemistry analysis on human healthy adult kidney for CDH6 protein, from human protein atlas (left). Immunofluorescence analysis on human healthy adult kidney for CDH6 (red) costained with CDH1 (green). (f) Immunofluorescence analysis on serial section of ADPKD kidney for VCAM1 (red, left) or CDH6 (purple, right) costained with CDH1 (green). Each pair of white numbers shared between left and right images indicates an identical cyst. VCAM1+CDH6+CDH- tubules/cyst indicate FR-PTC (upper; 3, 4) and VCAM1-CDH6+CDH- tubules indicate N-PTC (lower, 7). Scale bar indicates 50 μ m.

6- Interstitial cells and paracrine signaling: The authors suggest potential paracrine signaling of TNF α between collecting duct-derived cysts and interstitial ADPKD-specific fibroblasts (PKD-FIB). At the minimum, a ligand-receptor analysis including all cell types should be performed in order to rectify such a suggestion/hypothesis. This type of analysis will also be able to test the authors' prior claims/hypotheses of autocrine and paracrine signaling of FR-PTC as well as their later hypotheses of paracrine DHH signaling from arterial endothelial cells (AEC) to surrounding pericytes and fibroblasts.

[Response] We agree with the reviewer 2 that ligand receptor analysis would be needed for our hypotheses related to secretory factors in ADPKD microenvironment. We have outlined our response to each of the major criticisms below:

6A- Interstitial cells and paracrine signaling: The authors suggest potential paracrine signaling of TNF α between collecting duct-derived cysts and interstitial ADPKD-specific fibroblasts (PKD-FIB). At the minimum, a ligand-receptor analysis including all cell types should be performed in order to rectify such a suggestion/hypothesis.

[Response] We performed a ligand-receptor (LR) analysis including the fibroblast subtypes as well as all other cell types in ADPKD kidney dataset with CellChat in addition to previously mentioned New Fig. 4.

New Supplementary Figure 12 (b). Ligand-receptor analysis on all cell types and FIB subtypes for TNF signaling pathway: Circle plot showing an inferred network (left) or heat map showing interaction strength of signals from senders (secretors) to a receivers (targets) celltype (right) for TNF signaling pathway, among celltypes and FIB subtypes. Thickness of an arrow in a circle plot indicates interaction strength.

The LR analysis with CellChat suggests that TNF signaling from collecting duct-derived cyst cells to various cell types including PKD-FIB in ADPKD kidney.

6B- This type of analysis will also be able to test the authors' prior claims/hypotheses of autocrine and paracrine signaling of FR-PTC as well as their later hypotheses of paracrine DHH signaling from arterial endothelial cells (AEC) to surrounding pericytes and fibroblasts.

[Response] The LR analysis suggests that most of TGFβ signaling was sent from PT/FR-PTC in ADPKD kidneys, and the major targets cells were PT and distal nephron tubules (TAL and CNT_PC, **New Fig. 4**). The majority of the TGFβ signaling was predicted to be mediated by TGFB2, which was highly expressed in PT/FR-PTC (**New Supplementary Fig. 9**).

New Figure 4. Ligand-receptor analysis identified signaling network in ADPKD kidneys (a) Dot plot showing gene expression of *TGFB2* in each cell type in ADPKD or control kidneys. (b) Ligand-receptor analysis with CellChat. Circle plot showing an inferred network (left) or heat map showing interaction strength of signals from senders (secretors) to a receivers (targets) celltype (right) for TGFβ signaling pathway. Thickness of an arrow in a circle plot indicates interaction strength.

New Supplementary Figure 9. Bar plot showing relative contribution of each ligand-receptor pair in TGFβ signaling pathway.

Hedgehog (HH) signaling was predicted to be mainly from PT and TAL, and the main target was PT and TAL (**Fig. R7**). Furthermore, most of HH signals were mediated by SHH. These findings were not consistent with the notion that HH signal might be via DHH from endothelial subset (AEC) to fibroblasts as we had originally thought. We have removed the hypothesis from the manuscript, since our current data did not supply enough evidence to support the notion. Future high resolution spatial transcriptomic analysis may be useful to test the hypothesis (Please also see the response to the comment 7).

Figure R7. Ligand-receptor analysis on hedgehog signaling pathways in ADPKD kidneys
(a) Ligand-receptor analysis with CellChat. Circle plot showing an inferred network (left) or heat map showing communication probabilities from senders (secretors) to a receivers (targets) celltype (right) for hedgehog signaling pathway among all celltypes and FIB/ENDO subtypes. **(b)** Bar plot showing relative contribution of each ligand-receptor pair in hedgehog signaling pathway.

7- On a similar note, most of these speculative notions (which cell types might interact with one another in a paracrine fashion, as suggested both by their spatial proximity and their expression pattern of ligands and receptors) could be approached using spatial transcriptomics methods, which are available and have been used previously in the Humphreys lab. In this way, the proximity of FR-PTC, endothelial, interstitial, and other epithelial cells such as the collecting duct principal cells could be elucidated.

[Response] We agree that spatial transcriptomic approach will be useful to analyze cellular interactions. However, the resolution of currently available spatial transcriptomic approach in our lab (Visium, 10xGenomics) was not sufficient to suggest complicated ligand receptor interactions at a single cell level. Furthermore, necrotic nature with lower RNA quality of ADPKD samples may confound the ligand receptor analysis in spatial transcriptomic analysis. We think spatial transcriptomic approach is beyond the scope for this manuscript, but we hope to apply Visium HD spatial transcriptomic approach with single cell resolution, for ADPKD, after its release (expected early 2023). We thank the reviewer for the constructive suggestion, and we added discussion of this point as limitation in our manuscript (**page 11, line 44 - page 12, line 1**).

8- It is very confusing why the authors highlight GLII expression (which is hardly present in more than 2% of Myofib or PKD-FIB cells) rather than the more obvious finding from their dataset (IL6 expression in PKD-FIB, which was highly expressed in possibly 40-50% of cells, gauging from Suppl. Fig. 6b). The statement “GLII was highly expressed in FIB of ADPKD kidneys” seems overly optimistic to say the least, given the scarce positivity of cells in both Myo-FIB and PKD-FIB clusters (2% at the most!). The fact that the authors show the UMAP of all cells (Fig. 4j) rather than the subclustering UMAP for GLII expression (Suppl. Fig. 6d) in the main figure is misleading in this context. The dot plot visualizing avg. expression level and percentage of cells expressing the marker (Fig. 4g) should include all interstitial cells (including unknown1 and unknown2). From this plot, it is hard to gauge the fraction of cells expressing GLII, it might make sense to produce a separate dot plot because it looks like only ~1-2% of PKD-FIB cells are actually expressing GLII. Other than for the story connecting fibroblasts with Hedgehog pathway, it does not occur to me why the authors highlight this part of their data rather than the more striking and interesting finding from the subclustering analysis (IL6 expression in PKD-FIB cells). Most concerning to me is the fact that the low percentage of cells expressing GLII was not acknowledged in the manuscript text.

[Response] We agree on the whole. As mentioned above, we have concluded based on both reviewer’s comments that our prior analysis was insufficient to firmly implicate hedgehog signaling from endothelial subset to fibroblasts. For the reasons already enumerated above, we removed this hypothesis from our manuscript.

In contrast, the ligand receptor analysis among all celltypes (**New Fig. 4**) as well as that including FIB subtypes (**New Supplementary Fig. 12**) suggested that most of IL6 signaling was from FIB, especially PKD-FIB, and the major target cells were PT/FR-PTC in ADPKD kidneys. This finding was consistent with JAK-STAT pathway activation of FR-PTC (Fig. 5h). We have now addressed these points in the revised manuscript (page 6, line 47 - page7, line 4).

IL6 signaling pathway

New Supplementary Figure 12a. Ligand-receptor analysis on all cell types and FIB subtypes for IL6 signaling pathway: Circle plot showing an inferred network (left) or heat map showing communication probabilities of signals from senders (secretors) to a receivers (targets) celltype (right) for IL6 signaling pathway among celltypes and FIB subtypes. Thickness of an arrow in a circle plot indicates interaction strength.

9- Collecting duct cyst subtypes: Again, it looks as if the subclustering analysis has a substantial remaining batch effect, as evidenced by the substantial overlap between control and ADPKD cells before and hardly any overlap after subclustering, respectively (Fig. 5a). This might also explain the somewhat problematic trajectories with major discontinuities and noticeable gaps, again to be verified with other software packages and complemented by RNA velocity analysis.

[Response] The substantial overlap between CNT_PC clusters of control and ADPKD in the whole dataset UMAP is due to just relative closeness of gene expression patterns compared to other cell types. Unlike subclustering of PT/FR-PTC cluster in Fig.5, we observed sufficient overlap between ADPKD and control cells in subclustering analysis (**New Supplementary Fig. 13**), and we successfully detected ADPKD-specific cell state (PKD-CTC1, PKD-CTC2 and PKD-CTC3 [PKD-PC1, PKD-PC2 and PKD-PC3 in the original manuscript]).

New Supplementary Figure 13. Overlap between ADPKD and control cells in CNT_PC subclustering: (a) Proportion of disease group (control and ADPKD) in each subtype of subclustering on CNT_PC cluster. (b) UMAP colored by subtypes, split by individual samples.

We reformed trajectory analysis with Monocle2 or scVero on CNT_PC. Consistent with trajectory analyses on PT, the trajectories with Monocle2 showed less discontinuity of differentiation path, and trajectories with scVero seemed arbitrary (**Fig. R8**).

Figure R8. Trajectory inference of CNT_PC cluster (a) Pseudotemporal trajectories in subtypes of CNT_PC cluster in snRNA-seq were generated with Monocle2 with UMAP colored by pseudotime (left) or subtypes (right). (b) Trajectory inference was performed with scVero with UMAP colored by subtypes. The arrows indicate the direction of trajectories.

We have removed trajectory analysis on CNT_PC from the manuscript with the same reasons we mentioned above (please read the response to comment 3).

10- In the discussion, the authors suggest that “differences in responses to hypoxia may switch the trajectories to PKD-PC1 and PKD-PC2”, which seems to me very hypothetical, as the authors do not show any analyses in hypoxia and ischemia datasets that would indicate so. Again, the trajectories towards PKD-PC1 and PC2, respectively, seem problematic because of the substantial discontinuities, as outlined above.

[Response] We agree with the reviewer 2 that that statement was too hypothetical, and the trajectory analysis was problematic. We have now performed gene set enrichment analysis on subsets of CNT_PC cluster (**New Fig. 7d, e**).

New Figure 7. Activation of hypoxic response and glycolysis in PKD-CTC2 subtype

(d) Heatmap showing enrichment of hallmark gene sets in each cell type in CNT_PC clusters.

(e) UMAP displaying enrichment of genes up-regulated in response to hypoxia (upper) and genes encoding proteins involved in glycolysis and gluconeogenesis (lower) in snRNA-seq dataset.

Enrichment of genes up-regulated in response to hypoxia and genes encoding proteins involved in glycolysis were observed in PKD_CTC2 (PKD-PC2 in original manuscript) subtype in CNT_PC, suggesting that they are responding to hypoxia. As described in response to comment 3, the trajectory inference was not relevant in our dataset, and we do not have clues to determine if hypoxia induced PKD-CTC2 cell state or not. We have now addressed these points in the revised manuscript (**page 7, line 42 - 46**).

11A- The authors report differential accessibility for a distal enhancer of GPRC5A in seemingly PKD-specific collecting duct principal cells (PCs) and validate this by CRISPR interference not in PCs, but in proximal tubule cells (PTC), which is in itself strange and incoherent.

[Response] We have now repeated CRISPR interference (CRISPRi) on the 5' distal region or the promoter of *GPRC5A* in WT9-12 cells, an immortalized epithelial cell line from a renal cyst of an ADPKD patient. While CRISPRi on the promoter region achieved ~90% decrease of *GPRC5A* expression, targeting the 5' distal region induced a 40~50% decrease, confirming its enhancer activity (**New Fig. 8g**). CRISPRi on the promoter or the distal area unexpectedly slightly up-regulated the expression of surrounding genes in WT9-12 cells.

New Figure 8g. CRISPR interference on *GPRC5A* enhancer in cyst cell line.

RT and real-time PCR analysis of mRNAs for *GPRC5A* or its surrounding genes (*DDX47*, *HEBP1* and *GPRC5D*) in the WT9-12 cells with CRISPR interference targeting the promoter (Prom) or 5' distal potential enhancer (Enh) for *GPRC5A* gene. NT, non-targeting control. Each group consists of n = 6 data (2 sgRNAs with 3 biological replicates). Bar graphs represent the mean and error bars are the s.d. One-way ANOVA with post hoc Dunnett's multiple comparisons test.

The slight up-regulation of the surrounding genes may be due to the effect of the CRISPRi on the chromatin accessibilities of the surrounding gene loci or secondary effect of *GPRC5A* down-regulation, although these results still suggest that the enhancer activity of the 5' distal DAR is specific for *GPRC5A* gene (page 9, line 23 - 31).

We also repeated treatment of forskolin and ATRA on WT9-12 cells to evaluate the change in the expression of *GPRC5A* and replaced the data with that on primary RPTEC (New Fig. 8i).

New Figure 8i. *GPRC5A* expression is regulated by cAMP and retinoic acid signaling

RT and real-time PCR analysis of mRNAs for *GPRC5A* in WT9-12 cells treated with forskolin (10 μ M) with or without all-trans retinoic acid (ATRA, 1 μ M) for 6 h (n=3 biological replicates). Bar graphs represent the mean and error bars are the s.d. One-way ANOVA with post hoc Dunnett's multiple comparisons test.

11B -Can the authors verify the specificity of the CCAN for PCs? They should show corresponding gene browser views visualizing chromatin accessibility for all cell types (PCT, PST, FR-PTC, PEC_PODO,

TAL, DCT, CNT_PC, IC, ENDO, FIB, LEUK) with a corresponding CCAN. I am concerned that their results pertaining to chromatin accessibility of *GPRC5A* and its distal enhancer are not specific to collecting duct but also present in other cell types, namely PT cells.

[Response] We have not discussed the specificity of the CCAN around *GPRC5A* locus among celltypes in the original manuscript, since we utilized CCAN in ADPKD dataset for identification of potential enhancer generally related to *GPRC5A* gene expression in the original manuscript.

CCAN around *GPRC5A* gene in each celltype (**New Supplementary Fig. 18**) suggests that cis-coaccessibility between the 5' distal DAR and the promoter of *GPRC5A* was also detected in PCT+PST (PST was too small [96 cells] to predict CCAN, so PCT and PST were combined to analyze), FR-PTC, TAL and IC clusters as well as PKD-PC subset of CNT-PC. Detection of cis-coaccessibility in PCT/PST and FR-PTC despite a very small peak of the 5' distal DAR in these cell types (**New Supplementary Fig. 19a**) suggests robustness of the cis-coaccessibility between these two genomic regions. Detection of the CCAN in TAL and IC were consistent with mild up-regulation of *GPRC5A* gene expression among distal nephron cell types besides CNT_PC in ADPKD kidneys (**New Supplementary Fig. 19b**). Given the CRISPRi on the 5'distal DAR only down-regulated 40-50% of the amount of *GPRC5A* expression, there should be other mechanisms that also contribute to the considerable up-regulation of *GPRC5A* in PC. We have now discussed these points in the revised manuscript (**page 9, line 15 - 22**).

New Supplementary Figure 18. CCAN around *GPRC5A* locus in each cell type in ADPKD kidneys: Cis-coaccessibility between 5' distal enhancer (red boxes) and TSS of *GPRC5A* (blue column) in each celltype is shown.

New Supplementary Figure 19. Gene expression or chromatin accessibility of 5' distal enhancer for *GPRC5A* in each cell type of ADPKD kidneys: Accessibility on 5' distal enhancer of *GPRC5A* gene (red column, left) and dot plot showing *GPRC5A* expression in each cell type (right).

11c - Also, have the authors performed similar CRISPR interference assays for the distal enhancer element of *MIR31HG*?

[Response] We have now performed CRISPRi on the 5' distal DAR of *MIR31HG* in WT9-12 cells with newly designed three sgRNA targeting that regions (**New Supplementary Fig. 17**). CRISPRi achieved ~50% decrease of *MIR31HG* expression, confirming its enhancer activity to *MIR31HG*. *CDKN2A* expression was not changed in CRISPRi, suggesting that *CDKN2A* was not a direct target of that enhancer. *CDKN2A* expression was previously shown to be up-regulated by 70-80% reduction of *MIR31HG* expression by siRNA knockdown in human fibroblast cell line (Nat. Commun., 2021, Apr 28;12(1):2459). CRISPRi on the enhancer in WT9-12 cells induced ~50% reduction, which may be insufficient to induce gene expression change of *CDKN2A* in WT9-12 cells.

New Supplementary Figure 17. CRISPR interference on MIR31HG enhancer in cyst cell line.

RT and real-time PCR analysis of mRNAs for *GPRC5A* or *CDKN2A* in the WT9-12 cells with CRISPR interference targeting 5' distal potential enhancer (Enh) for *GPRC5A* gene. NT, non-targeting control. Each group consists of n = 6 (2 sgRNAs with 3 biological replicates for NT) or n=9 (3 sgRNAs with 3 biological replicates for Enh) data point. Bar graphs represent the mean and error bars are the s.d. Two-tailed Student's t-test.

Minor:

- What do the authors mean exactly when they say that increased promoter accessibility of *ADPKD* compared to control samples "... suggest[s] dynamic epigenetic remodeling"? This is somewhat of an imprecise statement. How do the authors explain this systematically higher promoter accessibility of *ADPKD* samples compared to control samples, which was true across all cell types?

[Response] We apologize for that imprecise statement. The number of DAR in each celltype of ADPKD was less than that of control, suggesting generally less chromatin accessibilities in ADPKD kidney cells. Given the lower quality of the snRNA-seq libraries from ADPKD samples, we think that the systematically lower DAR in the ADPKD samples may also reflect reduced chromatin quality. That said, the differences in the snATAC-seq were smaller in magnitude than those in the snRNA-seq datasets. This reduced chromatin quality may relatively enrich promoters which are more accessible than other genomic regions that were below the threshold of peak detection in the ADPKD dataset. The data showing promoter enrichment in DAR of ADPKD does not supply information regarding ADPKD mechanism, and the data would be confusing for the readers. We removed this data (original Fig. 2e) from the manuscript. Instead, we added the number of DAR in each celltype of control and ADPKD kidneys in the revised manuscript (**New Supplementary Fig. 8; page 5, line 5-8**)

Supplementary Figure 8. Number of DAR in each cell type of control or ADPKD snATAC-seq dataset: Bar graph showing the numbers of DARs in each celltype control (blue) and ADPKD (red).

Reviewers' Comments:

Reviewer #1:

Remarks to the Author:

The authors have been very responsive to the comments from the first review. They have not always been able to provide the requested data, such as the mutation data, but they have provided good explanations why this was not possible. In addition, some of the more speculative parts of the manuscript have now been removed. Overall, I am happy with the revisions that have been made.

Reviewer #2:

Remarks to the Author:

I have carefully reviewed the authors' revised manuscript and appreciate their focusing on a clearer storyline as well as their additional work on the 5' distal DAR of MIR31HG in a more appropriate cell line.

While I also appreciate the authors retracting their trajectory analyses, I believe that the inconsistencies associated with these data also highlight potential remaining issues surrounding the FR-PTC phenotype in the ADPKD samples: Especially the unbelievably high fraction of FR-PTC among ADPKD samples troubles me, and the authors fail to demonstrate coherent cell fractions with their staining data. Although the authors acknowledge that nuclei preparation bias in cystic tissue is a highly likely culprit of the loss of N-PTC cells, they should also acknowledge that the expression levels of VCAM1 (defining this population, after all) were lower in ADPKD samples compared to control samples, indicating a "dilution" effect, potentially due to mis-annotation of cells. Along those lines and given the relatively low log₂FC of both VCAM1 and CCL2 in FR-PTC compared to N-PTC (Suppl. Table 13), I am still convinced that a higher resolution clustering among PT cells would better differentiate healthy N-PTC and "true" FR-PTC and hence result in higher VCAM1 FR-PTC expression among ADPKD samples (by alleviating the dilution effect of true N-PTC currently annotated as FR-PTC). Also, to convincingly confirm their "FR-PTC" annotation, the authors should plot the correlation of averaged gene expression between their original study describing FR-PTC in IRI mice after orthologous human-mouse lift-over. Fig. 1c really should include VCAM1 and CCL2 expression as markers for FR-PTC to demonstrate accurate cell classification. Along those lines, the authors did not show double staining of VCAM1 and a PT marker (e.g., SLC34A1 or LRP2) to confirm VCAM1 expression in PT cells (LTL is expected to be decreased in damaged kidneys and therefore not suited for double-stainings).

Giving patient eGFRs as "<20ml/min" for all ADPKD samples neither differentiates inter-patient effects nor does it define end-stage kidney disease (defined as <15ml/min). Can the authors please look up patients' individual eGFR values? This might also elucidate whether the patient contributing PKD sample 3 (who contributed nearly all N-PTC cells from the ADPKD group, see Fig. 5b) had the highest eGFR.

Can the authors please make available the codes used for this study on GitHub, as indicated in the manuscript.

Reviewer #1 (Remarks to the Author):

The authors have been very responsive to the comments from the first review. They have not always been able to provide the requested data, such as the mutation data, but they have provided good explanations why this was not possible. In addition, some of the more speculative parts of the manuscript have now been removed. Overall, I am happy with the revisions that have been made.

[Response] We appreciate the positive evaluation of our revised manuscript.

Reviewer #2 (Remarks to the Author):

I have carefully reviewed the authors' revised manuscript and appreciate their focusing on a clearer storyline as well as their additional work on the 5' distal DAR of MIR31HG in a more appropriate cell line. While I also appreciate the authors retracting their trajectory analyses, I believe that the inconsistencies associated with these data also highlight potential remaining issues surrounding the FR-PTC phenotype in the ADPKD samples. Especially the unbelievably high fraction of FR-PTC among ADPKD samples troubles me, and the authors fail to demonstrate coherent cell fractions with their staining data.

[Response] We thank the reviewer for the careful evaluation of our revised manuscript and constructive critique. The central remaining concern is the possibility that the very high fraction of FR-PTC we observed in ADPKD samples is an artifact. We have carefully considered this possibility with new analyses and new experimentation, and we are convinced that our results are valid and also reflect the true state of PT in endstage ADPKD kidneys. In particular, we do not agree that in these ADPKD kidneys "there are scattered, albeit small, areas of relatively intact parenchyma...even in largely destroyed cystic kidneys." The implication is that these non-cystic tubules should be transcriptionally normal. But we have performed new immunofluorescence analyses which shows that many of these non-cystic tubules actually express VCAM1, a FR-PTC marker. These non-cystic tubules are surrounded by significantly expanded and fibrotic interstitium, and it stands to reason that the peritubular capillary network will be impaired as a consequence. We hypothesize that the very high FR-PTC fraction that we observe also reflects an injury response to hypoxia in these areas of "relatively intact parenchyma." We explore these issues in greater detail in response to the individual concerns below.

I - Although the authors acknowledge that nuclei preparation bias in cystic tissue is a highly likely culprit of the loss of N-PTC cells, they should also acknowledge that the expression levels of VCAM1 (defining this population, after all) were lower in ADPKD samples compared to control samples, indicating a "dilution" effect, potentially due to mis-annotation of cells.

[Response] We agree that the expression level of *VCAM1* in FR-PTC tends to be lower in ADPKD (average log2FC: 0.56) compared to that in control kidney (Fig. R1), although this finding was not statistically significant ($P = 0.15$, $P_{adj} = 1$). We have added this new data to the revised manuscript.

Figure R1. Violin plot showing VCAM1 expression in control or ADPKD FR-PTC cells in the original manuscript.

2 - Along those lines and given the relatively low \log_2FC of both *VCAM1* and *CCL2* in FR-PTC compared to N-PTC (Suppl. Table 13), I am still convinced that a higher resolution clustering among PT cells would better differentiate healthy N-PTC and "true" FR-PTC and hence result in higher *VCAM1* FR-PTC expression among ADPKD samples (by alleviating the dilution effect of true N-PTC currently annotated as FR-PTC).

[Response] We appreciate this point and agree that higher resolution clustering may identify additional PT subtypes, and that this may have been hampered by our prior approach lumping all PT nuclei together. To address this, we have now performed subclustering analysis on the ADPKD PT cells and FR-PTC (*VCAM1*+PT) in healthy kidneys (excluding the N-PT from healthy controls as suggested), and 4 clusters (PT-1/2/3/4) were identified (**New Fig. 5a, b**). The FR-PTC originating from control PT were clustered into PT-1/2 (47.1% and 49.3% of control cells, respectively). In contrast, almost all (> 99%) PT-3/4 were derived from ADPKD PT cells. PT-3/4 showed lower *VCAM1* expression compared to PT-1, although they expressed higher levels of other FR-PTC signature genes (*CREB5*, *TPM1*, *PROM1*[*CD133*], *TGFB2* and *HAVCR1*) despite lower *VCAM1* expression (**New Fig. 5c**), suggesting that *VCAM1* may not be a solo defining marker of the FR-PTC state. We have now included these points in the revised manuscript (**page 5, line 27 - 37; page 6, line 1 - 3**). Although beyond the scope of this manuscript, we are very interested in better defining and understanding heterogeneity amongst FR-PTC cell states, both in health and disease.

New Figure 5. (a) UMAP of healthy control PT lineage colored by subtypes (left and middle) or that displaying *VCAM1* expression (right). (b) Subclustering analysis on the ADPKD PT cells and FR-PTC (*VCAM1*+PT) in healthy kidneys. UMAP colored by diseases (left) or subtypes (middle). (c) Dot plot showing marker gene expressions among the PT subtypes in ADPKD kidneys. The diameter of the dot corresponds to the proportion of cells expressing the indicated gene and the density of the dot corresponds to average expression relative to all ADPKD PT cells.

3 - Also, to convincingly confirm their "FR-PTC" annotation, the authors should plot the correlation of averaged gene expression between their original study describing FR-PTC in IRI mice after orthologous human-mouse lift-over.

[Response] We compared the correlation of highly variable genes in the human and mouse IRI datasets for PT subsets (**New Fig. 5d**). The expression of mouse FR-PTC was more correlated with human FR-PTC in healthy kidneys or ADPKD PT subtypes compared to normal PT in control kidney. The correlation of gene expression between mouse FR-PTC and PT3 in ADPKD was weaker than other ADPKD subtypes, although the gene signature of PT3 was still closer to mouse FR-PTC than to other mouse PT subtypes. Together, the gene expression of ADPKD PT subtypes correlated well with FR-PTC in mouse IRI datasets. We have now included these points, data and methods in the revised manuscript (**page 5, line 38 - 44; page 17, line 21-28**).

d

New Figure 5 (d) Correlations of the averaged expressions of highly variable genes between PT subtypes in IRI mice and those of human dataset. The highly variable genes among IRI mice PT cells were identified with FindVariableFeatures function (nfeatures = 2,000) in Seurat, and the highly variable genes that also exist in human dataset after orthologous mouse-human lift over (biomaRt) were selected (1,648 genes). These highly variable genes were analyzed with Pearson correlation (cor function). The heat map shows Pearson correlation coefficients (R) between PT subtypes in IRI mice and those of human dataset (right).

4 - Fig. 1c really should include VCAMI and CCL2 expression as markers for FR-PTC to demonstrate accurate cell classification.

[Response] We added VCAMI and CCL2 in **New Fig. 1c**. The "PT" and "FR-PTC" annotation in the whole dataset are tentative ones, since more accurate annotation was done by sub-clustering analysis in Fig. 5. To avoid possible confusion to the readers, we changed the annotation of these clusters to PT1 and PT2.

New Figure 1 (c) Dot plot of snRNA-seq dataset showing gene expression patterns of cluster-enriched markers for ADPKD or control kidneys. The diameter of the dot corresponds to the proportion of cells expressing the indicated gene and the density of the dot corresponds to average expression relative to all cell types.

5 - Along those lines, the authors did not show double staining of *VCAM1* and a PT marker (e.g., *SLC34A1* or *LRP2*) to confirm *VCAM1* expression in PT cells (*LTL* is expected to be decreased in damaged kidneys and therefore not suited for double-stainings).

[Response] We performed new costaining for *VCAM1* with *CUBN* or *LRP2* in ADPKD samples (**New Fig. 5f**). In non-cystic tubules, we observed both *VCAM1*+*CUBN*+ as well as *VCAM1*+*CUBN*- cells in various frequencies dependent on areas observed in the tissues (**New Fig. 5e, f**). By contrast, nearly all *VCAM1*+ cells were also *LRP2*+, suggesting that *VCAM1*+ cells were of proximal tubular cell lineage. We also would like to emphasize that these *VCAM1*+, non-cystic tubules did not necessarily appear atrophic (**New Fig. 5f**). This observation is consistent with the notion that non-cystic ‘normal’ PT in ADPKD kidneys are in fact stressed/hypoxic and as a consequence have adopted a FR-PTC cell state. This would help to explain the very large fraction of FR-PTC in ADPKD kidneys. We have now included these points in the revised manuscript (**page 6, line 4 - 11**).

New Figure 5 (e) Violin plot showing *VCAM1*, *CUBN* or *LRP2* mRNA gene expression among PT subtypes of ADPKD kidney. **(f)** Immunohistochemistry analysis on human ADPKD kidney for *VCAM1* (green) and *CUBN* (red, left) or *LRP2* (red, right). Representative images of n=3 samples. Scale bar indicates 50 μm .

6 - Giving patient eGFRs as " $<20\text{ml/min}$ " for all ADPKD samples neither differentiates inter-patient effects nor does it define end-stage kidney disease (defined as $<15\text{ml/min}$). Can the authors please look up patients' individual eGFR values? This might also elucidate whether the patient contributing PKD sample 3 (who contributed nearly all N-PTC cells from the ADPKD group, see Fig. 5b) had the highest eGFR.

[Response] We added the individual eGFR at the time of nephrectomy to the Supplementary Table1. PKD sample 3 did not have the highest eGFR.

Sample ID	Gender	Age	eGFR (ml/min/1.73m ²)	Status prior to transplant	Duration of sample preservation (days)	Kidney size
PKD1	Female	56	7	Dialysis	810	1079 g
PKD2	Female	53	17	Pre-emptive	1207	803 g
PKD3	Female	58	17	Pre-emptive	216	1207 g
PKD4	Male	54	16	Pre-emptive	988	2061 g
PKD5	Female	61	23	Pre-emptive	535	1094 g
PKD6	Male	35	14	Pre-emptive	549	3170 g
PKD7	Male	42	10	Pre-emptive	629	2100 g
PKD8	Male	47	5	Pre-emptive	570	1531 g

7 - Can the authors please make available the codes used for this study on GitHub, as indicated in the manuscript.

[Response] We apologize that our GitHub was accidentally publicly unavailable. We have now made this publicly available. https://github.com/TheHumphreysLab/Multimodal_analysis_ADPKD

Reviewers' Comments:

Reviewer #2:

Remarks to the Author:

The authors have adequately addressed remaining concerns, especially those surrounding the FR-PTC cluster with additional analyses. I feel the discussion should reflect our remaining challenges in the field to better define heterogeneity amongst PT cell states in health and disease. In view of the sometimes not-so-great correlation of transcriptomic signatures between mouse and human (Fig. 5d), maybe the authors can also touch on the remaining difficulties and inconsistencies of annotating and naming those subclusters (e.g., FR-PTC, repairing, etc.).

Reviewer #2 (Remarks to the Author):

The authors have adequately addressed remaining concerns, especially those surrounding the FR-PTC cluster with additional analyses.

[Response] We thank the reviewer for the careful evaluation of our revised manuscript, and constructive critiques have been very helpful for improvement of our manuscript.

1 - I feel the discussion should reflect our remaining challenges in the field to better define heterogeneity amongst PT cell states in health and disease.

[Response] We appreciate this comment. We have now expanded our discussion of PT cell state heterogeneity in the discussion. Our subclustering analysis suggests previously unrecognized heterogeneity amongst FR-PTC cell states (Fig. 5b). Some PT subsets had lower *VCAMI* expression (PT-3/4) compared to the remaining (PT-1/2), although they expressed higher levels of other FR-PTC signature genes (Fig. 5c), suggesting that *VCAMI* is not a sole defining marker of the FR-PTC state, and that combinations of several markers may better classify damaged PT cell states. The heterogeneity of the failed repair state is reflected by the variability in correlation between ADPKD FR-PTC and mouse FR-PTC (Fig. 5d). We also described heterogeneity of proinflammatory or profibrotic signalling among these subsets (Fig. 5g, h), implicating a potentially unique role of each cell state in CKD or cyst progression. The extent to which each PT subset may contribute to disease progression remains undefined. A better understanding of PT heterogeneity and how these states contribute to disease remains a major future challenge. We have now included all of these points in the revised discussion section.

2 - In view of the sometimes not-so-great correlation of transcriptomic signatures between mouse and human (Fig. 5d), maybe the authors can also touch on the remaining difficulties and inconsistencies of annotating and naming those subclusters (e.g., FR-PTC, repairing, etc.).

We agree that the reduced correlation of transcriptomic signatures between mouse FR-PTC and especially the PT-3 subset (Fig. 5d) poses a question if this subset can be annotated as a true FR-PTC. PT-3 strongly expressed some FR-PTC marker genes (*CREB5*, *TPM1*, *PROM1*[*CD133*], *TGFB2*) but did not express *VCAMI*, suggesting that this cell state may be transitioning either toward or away from *VCAMI*+ cell states. We have now explicitly acknowledged these points in the revised manuscript (**page 10, line 14 - 16**).